



## Assessment of Integrated Watershed Health based on Natural Environment,

## Hydrology, Water Quality, and Aquatic Ecology

So Ra Ahn[a] and Seong Joon Kim[b]
[a]Assistant Research Scientist (Ahn), Texas A&M AgriLife Research Center at El Paso, Texas 79927, USA; and
[b]Professor (Kim), Department of Civil and Environmental System Engineering, Konkuk University, Seoul 05029,
South Korea, Email: kimsj@konkuk.ac.kr

11                                                    **Abstract**

Watershed health, including the natural environment, hydrology, water quality, and aquatic ecology, was assessed for
the Han River basin (34,148 km²) in South Korea using the Soil and Water Assessment Tool (SWAT). The evaluation
procedures followed those of the Healthy Watersheds Assessment by the U.S. Environmental Protection Agency (EPA).
To evaluate watershed health (basin natural capacity), 6 components of the watershed landscape were examined:
stream geomorphology, hydrology, water quality, aquatic habitat condition, and biological condition. In particular, for
the hydrology and water quality components, the SWAT was applied for the study basin with 237 sub-watersheds
(within a standard watershed on the Korea Hydrologic Unit Map) and including three multipurpose dams, one
hydroelectric dam, and three multifunction weirs. The SWAT was calibrated (2005–2009) and validated (2010–2014)
using each dam and weir operation, the flux tower evapotranspiration, TDR soil moisture, and groundwater level data
for the hydrology assessment and using sediment, total phosphorus, and total nitrogen data for the water quality
assessment. The water balance considering the surface–groundwater interactions and the variation in stream water
quality were quantified according to the sub-watershed-scale relationship between the watershed hydrologic cycle and
stream water quality. We assessed the integrated watershed health according to the U.S. EPA evaluation process based
on the vulnerability levels of the natural environment, water resources, water quality, and ecosystem components. The
results suggest that approaches aimed at simultaneously improving the water quality, hydrology, and aquatic ecology
conditions may be necessary to improve integrated watershed health.





Keywords: Watershed health assessment; SWAT; Watershed hydrology; Water quality; Aquatic ecology





**1. Introduction**
Watershed management can be defined as the integrated and iterative decision process applied to maintain the
sustainability of resources through the balanced use and conservation of water quantity, land, vegetation, and other
natural resources within the watershed. The river is a constituent element of the watershed ecosystem that is of primary
concern for watershed management; the river discharge and water quality are key components of the watershed
ecosystem, and their interactions can be affected by land use and vegetation cover. The Han River basin in South
Korea, with its large-scale water supply dams and weirs, is a rare case worldwide. Twenty-six years ago, the
government initiated programs designed to restore the environmental and human health-related quality of the Han
River basin. However, an integrated approach considering water supply, water quality improvement, and natural
ecosystem maintenance and their interactions within the watershed was lacking. It has become clear that a broader
view of watershed ecosystems is essential if we are to truly protect the chemical, physical, and biological integrity of
our watersheds (U.S. EPA, 2012).
One of the key components of watershed management strategies is to increase the protection of healthy waters,
including healthy watersheds. A key component of watershed health is its ability to withstand, recover from, or adapt
to disturbances, such as floods and droughts. A more complete understanding of the watershed ecosystem components
affecting watershed health is important for identifying management actions to protect healthy watersheds. Without an
integrated watershed health assessment system, the successes in restoring impaired waters will be limited and the
many socioeconomic benefits of healthy watershed systems will be lost.
In general, the assessment of the major components of watershed health must incorporate evaluations of the natural
environment, hydrology, water quality and aquatic ecology. A number of studies have recently assessed the potential
for effective watershed management through an analysis of a variety of health indicators. Sanchez et al. (2015)
characterized the relationships between in-stream health indicators (flow, sediment, and nutrient loads) using the Soil
and Water Assessment Tool (SWAT) model and socioeconomic measures of communities using spatial clustering
techniques and confirmatory factor analysis in the Saginaw River watershed in Michigan. Cook et al. (2015) explored
the effects of both water quality and habitat on benthic macroinvertebrates using the data from a three-year field study
and Virginia Stream Condition Index (VSCI) scores to evaluate site-specific environmental variables (land use, habitat
metrics, water quality parameters), examining these relationships in five watersheds along the Virginia–Kentucky
border. Tango and Batiuk (2016) analyzed interactions affecting the watershed and bay water quality recovery





responses to management actions and a range of health conditions and impairments by measuring the physical,
chemical and biological parameters in Chesapeake Bay.
The U.S. EPA has made considerable efforts to move towards integrated evaluations of watershed health. For
example, The Virginia Watershed Integrity Model uses an integrated approach to evaluate the landscape condition and
terrestrial habitat to identify ecologically important catchments across the landscape (Virginia Department of
Conservation and Recreation, 2008). Minnesota's Watershed Assessment Tool used hydrology, geomorphology,
biology, connectivity, and water quality data in an integrated context to evaluate the health of Minnesota's watersheds
(Minnesota Department of Natural Resources, 2011). The Oregon Watershed Assessment addressed landscape, habitat,
biology, water quality, hydrology, and geomorphology through field assessments and follow-up analyses based on a
classification and condition assessment of channel habitat types (Watershed Professionals Network, 1999). The
California Watershed Assessment Manual evaluated the six essential ecological attributes of landscape status:
hydrology/geomorphology, biotic condition, chemical/physical condition, natural disturbance regimes, and ecological
condition (Shilling, 2007).
Regional water quantity and quality can be assessed by systematic modeling using the hydrologic model SWAT
(Arnold et al., 1998) because of its robust approach based on the soil water balance at the watershed scale. The SWAT
model has been successfully applied to a number of river basins and is widely used to study the long-term impacts of
hydrological (e.g., Sun and Cornish 2005; Wan et al., 2013; Ahn et al., 2016; Karlsson et al., 2016; Sellami et al., 2016;
Chung et al., 2017) and environmental changes (e.g., Eckhardt and Ulbrich, 2003; Rosenberg et al., 2003; Bouraoui
et al., 2004; Chaplot, 2007; Mehdi et al., 2015; Zhou and Li, 2015). Thus, the use of this qualified watershed model is
highly useful for assessments of continuous time-series changes and spatial distributions changes in watershed
information.
However, most previous studies have employed a fragmentary approach to investigating one or several
environmental issues using monitoring data for a limited period without assessing the various components (e.g.,
landscape, stream channels, hydrology, water quality, habitat, biological diversity, etc.). Thus, the methodology
suggested in this study is essential to explore the integrated influence of large-scale watersheds with various watershed
characteristics and to assess the overall health of watersheds.
Therefore, the main objective of this study is to conduct a watershed health assessment analysis of the natural
environment, hydrology, water quality, and aquatic ecology of the Han River basin (34,148 km²) in South Korea using





monitoring data and SWAT modeling output. Detailed information regarding the framework is presented below.

## 2. Materials and methods

2.1 Methodology for watershed health assessment
The foundation of watershed health assessment is the compilation and summarization of watershed parameters based
on the primary physical attributes of watershed conditions. According to the United States Environmental Protection
Agency (U.S. EPA, 2012), there are six essential indicators fundamental to the assessment of watershed health: 1)
landscape condition, 2) geomorphology, 3) hydrology, 4) water quality, 5) habitat, and 6) biological condition. A sub-
index for each of the six components is developed from these indicators. The sub-index values are then aggregated
into a single Watershed Health Index value for each watershed. This methodology can be used to assess the natural
capacity of a watershed and its problems and to draft possible solutions for effective watershed management. All sub-
index and index values are relative (i.e., "healthier" vs. "not as healthy") rather than absolute (i.e., no "healthy vs.
unhealthy" cutoff score is identified) and thus are meant for comparing the relative differences among watersheds
rather than precisely defining healthy vs. unhealthy watersheds.
In this study, the indicators for watershed health assessment are selected based on the six essential components and
methodology suggested by the U.S. EPA. All of the indicators for watershed health are evaluated to match the situation
in South Korea using measurable data or watershed modeling results. In particular, the methodology is developed to
assess the effects of hydrology and water quality on watershed health to analyze the possible long-term changes in the
watershed as simulated through a watershed-scale hydrological model, the SWAT. According to existing research that
has assessed the long-term changes in the Han River basin, the changes in runoff due to climate change in Han River
basin is expected to cause many changes to the future seasonal water volume, and water scarcity is predicted to increase
in the long term (Jun et al., 2011; Kim et al., 2014). Urban land cover in the Han River basin is positively associated
with increases in water pollution, which has increased for the majority of the monitoring stations (Chang, 2008).
Healthy areas can be identified based on standard watersheds from Korea Hydrologic Unit Map. The Korea
Hydrologic Unit Map is a standard map that combines data from national organizations for water resource
development, planning, and management. The standard watersheds are the smallest hydrologic unit designated by the
Korean government. Figure 1 shows a flowchart of the modeling procedures. The specific objectives of this study are
as follows:





- To develop a method for reconstructing water quantity and quality time-series data of the basin using the SWAT model. The reconstructed time-series are used as water quantity and quality indicators and for sub-index development. Because watershed health assessment relies on the continuous flow of time-series information, the SWAT model was established and calibrated to obtain flow records at ungauged hydrology and water quality stations.

- To establish a reference condition for each indicator to assess the sub-index through normalization of the following components: landscape condition, geomorphology, hydrology, water quality, habitat, and biological condition.

- To assign integrated watershed health scores combining multiple indicators representing different attributes of healthy watersheds based on a standard watershed on the Korea Hydrologic Unit Map.

<Figure 1>

2.2 Study area description

The Han River basin (34,148 km²) is one of the five major river basins in South Korea (99,720 km²); it occupies approximately 31% of the country and falls within the latitude-longitude range of 36.03° N to 38.55° N and 126.24° E to 129.02° E, respectively (Figure 2). The basin has three main rivers, the North Han River (12,969 km²), the South Han River (12,894 km²), and the Imjin River (8,285 km²). The North Han River and South River merge and then flow into the metropolitan city of Seoul, a city of 10 million residents. The water resources of the river basin must be managed sustainably due to the expanding water demand of the Seoul area, including its satellite cities (12 million individuals), and the potential changes to water resources due to climate change must be evaluated (Ahn and Kim, 2016). The dominant land use of the Han River basin is forest (73%, 25,033 km²), followed by cultivated cropland in the lowland fertile areas (5,915 km²), including rice paddy fields (6%) and upland crops (12%) (Figure 2b). Over the 30 years of weather data from 1985 to 2014, the average annual precipitation is 1,395 mm and the annual mean temperature is 11.5 °C. Figure 2a shows the study area and the 237 sub-watersheds (within a standard watershed on the Korea Hydrologic Unit Map) delineated for the SWAT modeling and watershed health assessment, and Figure 2c shows the four test areas for comparison of the watershed health index scores.



<Figure 2>

2.3 Data collection
A summary of datasets and associated organization sources, metrics, and measurement methods used in the assessment
is provided in Table 1. These data were used to calculate the health assessment components for each of the six
watersheds.

For the landscape, stream geomorphology and aquatic habitat assessment, Geographic Information System (GIS)

datasets were used. The elevation data used the 90 m grid size Shuttle Radar Topography Mission (SRTM) digital
elevation model (DEM) supplied by the International Center for Tropical Agriculture (CIAT). The land cover map for
nine classes of land cover (coniferous forest, deciduous forest, mixed forest, paddy rice, upland crop, urban, grassland,
bare field, and water) for 2008 was obtained from the Korea Ministry of Environment (KME). The stream map for
national and local streams was obtained from the Ministry of Land, Infrastructure, and Transport (MOLIT) of South
Korea. The information on the location and number reservoirs for the Han River basin was obtained from the Korea
Rural Community Corporation (KRC).

For the hydrology and water quality assessments, the SWAT modeling outputs for a total of 237 sub-watersheds

for the Han River basin, including ungauged locations, were used. The monitoring data for hydrology include only
streamflow and do not include data for the water balance components associated with the surface–groundwater
interaction. The monitoring data for water quality are not exhaustive. The period of the water quality components of
interest for this study, such as sediments, total nitrogen (T-N) and total phosphorus (T-P), is not sufficient to analyze
long-term changes. The daily continuous record of precipitation (PREC), total runoff (TQ), surface runoff (SQ),
infiltration (INFILT), soil water storage (SW), lateral flow (LQ), percolation (PERCOL), groundwater recharge
(RECHARGE), and return flow (GWQ) data for the hydrology metric and sediment, T-N, and T-P for the water quality
metric were obtained from SWAT modeling for a thirty-year period (1985–2014).

For the biological assessment, the monitoring data were obtained from the Korea Ministry of Environment (KME)

in South Korea which has been monitoring river ecological data for 360 monitoring stations in the Han River and its
tributaries since 2008. Samples of trophic diatom communities (339 species), benthic macroinvertebrate communities
(344 species), and fish communities (394 species) were collected from the monitoring stations in September and
October of each year during the six years (2008–2013) and the Trophic Diatom Index (TDI), Benthic





Macroinvertebrate Index (BMI), and Fish Assessment Index (FAI) were calculated and classified by ranking the
arithmetic means. Details of the data collection and calculation procedures are provided in the Nationwide Aquatic
Ecological Monitoring Program Report (Ministry of Environment, 2013).

<Table 1>

2.4 Hydrology and water quality simulations using the SWAT model

The SWAT model is a physically based, continuous, long-term, distributed parameter model designed to predict the
effects of land management practices on hydrology and water quality in agricultural watersheds under varying soil,
land use, and management conditions (Arnold et al., 1998). The SWAT model is based on the concept of hydrologic
response units (HRUs), which are portions of a sub-basin with unique land use, management, and soil attributes. The
runoff, sediment, and nutrient loadings from each HRU are calculated separately based on weather, soil properties,
topography, vegetation, and land management and are then summed to determine the total loading from the sub-basin
(Neitsch et al., 2002). A detailed description can be found in the Soil and Water Assessment Tool user's manual and
theoretical documentation (Neitsch et al., 2005).
The watershed health assessment requires the indicator data for hydrology and water quality to be simulated by the
SWAT model, and the detailed component selection is presented in Sections 2.5.3 and 2.5.4. This section briefly
summarizes the model data and implementation and the statistical results of calibration and validation.

2.4.1 Measured data for the SWAT model evaluation
The Han River Basin was divided into 237 sub-watersheds and 1,987 HRUs for SWAT modeling. The sub-watershed
delineation was defined using the 90 m SRTM DEM supplied by the CIAT. A 2008 land cover map for nine classes
(coniferous forest, deciduous forest, mixed forest, paddy rice, upland crop, urban, grassland, bare field, and water)
were obtained from KME (Figure 2b). A soil map containing texture, depth and drainage attributes was rasterized to
a 90 m grid size from a 1:25,000 scale vector map supplied by the Korea Rural Development Administration (RDA)

In this study, three multipurpose dams (Hoengseong, Soyang, and Chungju), one hydroelectric dam (Paldang), and

three multifunction weirs (Kangcheon, Yeoju and Ipo) were selected as SWAT model calibration points (Figure 2a).





The Hoengseong Dam (HSD) and Chungju Dam (CJD), located in the upstream region of the South Han River basin,
have storage capacities of 87 million m³ and 2.8 billion m³, respectively. Its storage capacity makes CJD the second
largest dam in South Korea. The Soyang Dam (SYD), located upstream in the North Han River basin, has a storage
capacity of 2.9 billion m³, making it the largest dam in South Korea. The Kangcheon weir (KCW), Yeoju weir (YJW)
and Ipo weir (IPW) were constructed by the government in 2012 to secure water resources and prevent flooding. These
weirs are directly linked to the Paldang Dam (PDD), which can supply more than 2.6 million m³ of water per day to
Seoul and its metropolitan areas and has a storage capacity of 244 million m³. The observation data were prepared to
evaluate the SWAT model and simulate of the hydrological cycle and water quality including daily meteorological
data, dam inflow, dam outflow, dam storage, evapotranspiration, soil moisture, sediments, T-N, and T-P. Thirty-one
years (1984–2014) of daily meteorological data (precipitation, maximum and minimum temperature, relative humidity,
wind speed, and solar radiation) were collected from nineteen weather stations of the KMA. For the calibration and
validation of the watershed hydrology with dam operations, ten years (2005–2014) of daily dam inflow, outflow and
storage volume data for the multipurpose dams were obtained from three water level stations (HSD, SYD and CJD)
monitored by the Korea Water Resources Corporation and one water level station (PDD) monitored by the Korea
Hydro & Nuclear Power Co., Ltd. In addition, two years (2013–2014) of daily measured dam inflow, outflow and
storage volume data for the three multifunction weirs (KCW, YJW and IPW) monitored by the Korea Water Resources
Corporation were used. For the calibration and validation of stream water quality, ten years (2005–2014) of eight-day
intervals for sediments, T-N, and T-P data were obtained from seven stations (SG, CSG, JW, KCW, YJW, IPW, and
PDD) for the hydrology monitored by the KME. Figure 2a shows the gauging stations for the SWAT modeling.

2.4.2 Calibration and validation of the model

The SWAT model was calibrated at seven locations in the main river reaches using five years (2005–2009) of daily

inflow, storage volume data for the dams and weirs, sediments, T-N, and T-P data and was subsequently validated
using another five years (2010–2014) of data using the average calibrated parameters. In addition, the model was
spatially calibrated and validated using evapotranspiration and soil moisture data measured at two locations (SM and
CM) and groundwater level data measured at five locations (GPGP, YPGG, YPYD, YIMP, and HCGD) over five years
(2009–2013). The parameters were calibrated by trial and error until they achieved the necessary modeling
performance. The calibrated parameters and hydrograph of the calibration results in the Han River basin were





described by Chung et al (2017).
The statistical results for hydrology and water quality for the model calibration and validation are summarized in
Table 2. The coefficient of determination ($R^2$), the Nash and Sutcliffe model efficiency (NSE), the root-mean-square
error (RMSE), and the percent bias (PBIAS) were used to evaluate the ability of the SWAT model to replicate temporal
trends in the observed hydrological and water quality data. In the case of dam inflow, the $R^2$ value was greater than
0.59. The average NSE was 0.59 at HSD, 0.78 at SYD, 0.61 at CJD, 0.79 at KCW, 0.77 at YJW, 0.88 at IPW, and 0.87
at PDD. The PBIAS values of HSD, CJD, SYD, KCW, YJW, IPW and PDD were 13.5%, 12.2%, 9.4%, 11.5%, 19.8%,
21.4%, and 4.5%, respectively. In the case of the dam storage volume, the average $R^2$ was between 0.40 and 0.96 and
the PBIAS was between 0.9% and 18.9% for each calibration point. The average $R^2$ for evapotranspiration was
between 0.70 and 0.81, the soil moisture was between 0.75 and 0.85, and the groundwater level was between 0.40 and
0.70 for each calibration point. The average $R^2$ for the sediment was between 0.54 and 0.90, T-N was between 0.46and
0.82, and T-P was between 0.47 and 0.80 for each calibration point. The calibration results were consistent with the
SWAT calibration guidelines (NSE$\geq$0.5, PBIAS$\leq$28%, and $R^2\geq$0.6, Moriasi et al., 2007; Santhi et al., 2001) and were
found to be satisfactory.

<Table 2>

2.5 Data reconstruction for watershed health assessment
2.5.1 Landscape condition
The area of natural land cover (forest, wetland, river, and natural grassland) within a watershed can be an important
indicator of watershed health. Impervious land cover associated with roads and residential and urban areas can increase
watershed runoff, leading to instream flow alteration, geomorphic instability, and increased pollutant loading.
According to previous studies, a smaller area of impervious land cover may have significant impacts on aquatic
ecosystem health (e.g., King et al., 2011; Wang and Yin, 1997).
The extent and connectivity of the natural land cover within a watershed are very important for ecological integrity.
Natural land cover within the watershed, and especially within headwater areas and riparian corridors, helps to
maintain the hydrologic regime, regulates inputs of nutrients and organic matter, and provides habitat for fish and
wildlife (U.S. EPA, 2012). In the present study, assessing the connectivity of the natural land cover (forest, wetland,



river, and natural grassland) of watersheds involved a green area assessment; green areas comprise areas of
unfragmented natural land cover and corridors of sufficient width to allow the migration of wildlife between the
watersheds (Figure 3a). For the 237 sub-watersheds of Han River basin, the percentage of each watershed area
occupied by natural land cover (habitat blocks) was calculated using GIS techniques. The green area metric was
calculated as follows:

$Green\ area\ metric = \frac{Area\ (km^2)\ of\ natural\ land\ cover\ in\ watershed}{Total\ area\ (km^2)\ in\ watershed}$     (1)

The amount of natural land cover within the active river area is another important indicator of the landscape
condition. The natural land cover within the active river area, including the river channel, lakes and ponds, and the
riparian lands, is necessary for the physical and ecological functioning of the aquatic ecosystem (U.S. EPA, 2012).
Active river areas, in their natural state, maintain the ecological integrity of rivers, streams, and riparian areas and the
connection of those areas to the local ground water system (IPCC, 2007). The methods used to delineate the active
river area involve GIS techniques and analyses of elevation, land cover, and wetlands data. For the streamside areas
not yet decided the criteria for identifying, an area with a width of 30–50 meters can be used as a cutoff for identifying
streamside material contribution areas (US. EPA, 2012). In this study, for the 237 sub-watersheds of Han River basin,
the percentage of natural land cover within the riparian area within 50 meters of stream was calculated for each
watershed using GIS techniques (Figure 3b). The active river area metric was calculated as follows:

$Active\ river\ area\ metric = \frac{Area\ (km^2)\ of\ natural\ land\ cover\ in\ active\ river\ area}{Total\ area\ (km^2)\ in\ active\ river\ area}$     (2)

<Figure 3>

2.5.2 Stream geomorphic condition
The natural stream geomorphology can be an important indicator of watershed health because it can fragment both
the terrestrial and aquatic habitats throughout a watershed. Kline et al. (2009) performed detailed assessments of
stream geomorphic conditions using the Vermont Stream Geomorphic Assessment Protocols for the streams of



Vermont, USA. The assessment protocols are GIS-based analyses using elevation, land cover, and stream network
data layers to classify stream types and evaluate the conditions of individual reaches based on a comparison to
reference conditions for that stream type.
Table 3 provides descriptions of the stream geomorphic condition categories that are determined through the stream
impact rating and the stream order for the watershed health assessment of the geomorphic condition in the Han River
basin. In this study, the assessment of geomorphic condition was performed in a manner similar to that used for the
stream condition categories of the Vermont Stream Geomorphic Assessment Protocols. The stream order was
calculated for nine levels (Figure 4a) using a DEM and stream map, and four river classifications were created through
follow-up with detailed land cover assessments (Figure 4b). There are four river classifications for reference
(mountainous river, stream order 1), good (small river, stream orders 2–3), fair (local river, stream orders 4–5), and
poor (urban and national river, stream orders 6–9). The percentage of the assessed stream length in the reference
condition was calculated for each watershed. The stream geomorphology metric was calculated as follows:

$$Stream\ geomorphology\ metric = \frac{Stream\ length\ (km)\ of\ reference\ condition\ in\ watershed}{Total\ stream\ length\ (km)\ in\ watershed} \qquad (3)$$

<Figure 4>
<Table 3>

2.5.3 Hydrologic condition
The assessment of the hydrologic condition of a watershed requires long-term streamflow observation data for the 237
sub-watersheds of Han River basin. However, there were not enough gauging stations to fully assess the entire
watershed over the full thirty-year period. There were no data for the water balance components associated with the
surface–groundwater interaction, except for streamflow. Where long-term flow data are not available, they can be
estimated using hydrologic modeling techniques. To this end, the SWAT hydrologic model was used to simulate the
water balance components within the Han River basin.
To simulate the potentially available water quantity of the basin, the model was applied by dividing the basin into
237 sub-watersheds considering the water resources facilities operation (inflow and storage volume) of three
multipurpose dams, one hydroelectric dam, and three multifunction weirs. The SWAT simulation outputs—including



PREC and TQ for the total processes; SQ for the surface processes; INFILT, SW, and LQ for the soil water dynamics;
and PERCOL, RECHARGE, and GWQ for the groundwater dynamics—of each of the 237 sub-watersheds were
reported. All the results of the SWAT model were output in mm.

313        The annual average water balance components at the surface, in the unsaturated zone, and in a shallow aquifer can

serve as indicators of potential hydrologic alteration. The surface water and lateral groundwater flow interactions were
of major importance for the water balance in the Han River basin. In particular, infiltration, return flow, and
groundwater recharge were important factors for the whole hydrological cycle. In this study, the SWAT model results
were used to reconstruct daily time-series for the hydrologic components PREC, TQ, SQ, INFILT, SW, LQ, PERCOL,
RECHARGE, and GWQ for a thirty-year period (1985–2014) (Figure 5). The annual average value for the total of the
237 sub-watersheds during this period was used as the reference condition (Table 4). Dividing the simulated value of
the watershed by the reference condition yields the storage ratio of the nine components. The storage ratios of the nine
components were divided into four hydrologic classifications—total metric (PREC and TQ), surface processes metric
(SQ), soil water dynamics metric (INFILT, SW, and LQ), and groundwater dynamics metric (PERCOL, RECHARGE,
and GWQ)—for use in establishing specific management objectives. The storage ratio of each component for the four
hydrology metrics was calculated for each watershed and used as a metric of the hydrologic condition. The hydrology
metric was calculated as follows:

$$Hydrology\ metric = \frac{Simulated\ value\ (mm)\ (PREC,TQ,SQ,INFILT,SW,LQ,PERCOL,RECHARGE,and\ GWQ)\ of\ watershed}{Average\ value\ (mm)\ for\ all\ watersheds\ in\ basin} \qquad (4)$$

<Figure 5>

2.5.4 Water quality condition
The assessment of the water quality of the watershed also requires long-term observational data from the 237 sub-
watersheds of the Han River basin. However, the monitoring data for water quality are not exhaustive and not sufficient
to analyze long-term changes. In this study, the SWAT model was used to simulate the water quality sediment loads
(tons), T-N (kg) and the T-P (kg) within the Han River basin.

The SWAT model results were used to reconstruct load-based daily time-series for the water quality constituents

sediment (mg/L), T-N (mg/L), and T-P (mg/L) for a thirty-year period (1985–2014) (Figure 6). As part of the Basic





Environmental Policy Act (BEPA), South Korea has specified ecoregional water quality criteria for identifying the
least-disturbed sites throughout South Korea. These criteria were used to identify the streams and lakes that are likely
to be in the reference condition based on their sediment, T-N, and T-P concentrations. The "marginally good" level of
a seven-point scale (excellent, very good, good, marginally good, fair, poor, very poor) of water quality criteria for
streams and lakes was used for the reference condition (Table 4). The percentage of the assessed value in the reference
condition was calculated for each watershed. The water quality metric was calculated as follows:

$$Water\ quality\ metric = \frac{Simulated\ value\ (mg/L)\ (sediment,\ T\text{-}N,\ and\ T\text{-}P)\ of\ watershed}{Reference\ value\ (mg/L)\ in\ watershed} \qquad (5)$$

<Figure 6>

2.5.5 Aquatic habitat condition
The quality of aquatic habitat is dependent on the surrounding landscape and the hydrologic and geomorphic processes.
Therefore, habitat condition is partly accounted for through indicators representing those assessment components. The
potential for organisms to migrate upstream and downstream within a riverine system can also serve as an indicator
of aquatic habitat condition. Lakeshores also have riparian zones that serve as a source of organic material to the lake
aquatic habitat and help stabilize the lake perimeter (U.S. EPA, 2012). EPA's National Lakes Assessment (NLA)
identified poor lakeshore habitat as the most prominent stressor to the biological health of lakes (U.S. EPA, 2009).
The density of reservoirs per stream length was calculated and used as an indicator of aquatic habitat connectivity
(Figure 7a). The aquatic habitat connectivity metric was calculated as follows:

$$Aquatic\ habitat\ connectivity\ metric = \frac{Number\ of\ reservoirs\ in\ watershed}{Total\ stream\ length\ (km)\ in\ watershed} \qquad (6)$$

Intact wetlands help to maintain natural hydrologic regimes, provide important habitat for fish and wildlife, and

regulate water quality. The percentage of the watershed occupied by wetlands was calculated and used as an additional
indicator of habitat condition for each watershed (Figure 7b). The wetland metric was calculated as follows:





$$Wetland\ metric = \frac{Area\ (km^2)\ of\ wetlands\ in\ watershed}{Total\ area\ (km^2)\ in\ watershed} \qquad (7)$$

<Figure 7>

2.5.6 Biological condition
Based on the understanding that aquatic ecological environmental degradation is one of the leading causes of stream
impairment, the Ministry of Environment of South Korea began collecting variables of biological community diversity
as part of its Nationwide Aquatic Ecological Monitoring Program for a six-year period (2008–2013). Based on a
statistical evaluation of these data, three biological indicators (TDI, BMI, and FAI) were chosen to identify healthy
instream conditions for the Han River basin. In the Han River basin, the TDI, BMI, and FAI were developed from
epilithic diatoms, benthic macroinvertebrates, and fish assessments for estimating the overall biological condition
during the six years (2008–2013); these data can be used to identify healthy instream conditions in the context of
aquatic ecosystem health. Healthy watersheds should have TDI, BMI, and FAI scores close to the reference conditions.
According to the Nationwide Aquatic Ecological Monitoring Program Report (Ministry of Environment, 2013), the
indices with a range from 0 to 100 were classified on a four-point scale of best, good, fair, and poor for the biological
condition criteria, and the best and good levels were used as the reference condition (Table 3). The percentage of the
assessed scores on the TDI, BMI, and FAI in the reference condition was calculated for each watershed (Figure 8).
The biological condition metric was calculated as follows:

$$Biological\ condition\ metric = \frac{Observed\ value\ (TDI, BMI, and\ FAI)\ of\ watershed}{Reference\ value\ for\ watershed} \qquad (8)$$

<Figure 8>

2.6 Watershed health index formulation
The definition of the watershed health index is presented by the U.S. EPA for integrated watershed health evaluations.
Watershed health was evaluated by normalizing the metric scores to integrate the data on multiple healthy watershed
attributes into a composite score. Normalization was conducted by simply defining a reference value for the indicator





score that is considered healthy based on percentile rank. For communication purposes, the indicator score was scaled
to normalize the final sub-index and watershed health index scores to range from 0 to 1. Table 4 shows the definition
of the "healthy" reference value for the hydrology, water quality, and biological indicators. The indicator scores must
also be directionally aligned, meaning that higher scores should equate to "better" conditions for each metric. For
metrics that are not directionally aligned in their original units (e.g., water quality components), the inverse (1/X) of
each value can be taken.
A composite index of watershed health was constructed by averaging the normalized indicator scores for each
attribute. For attributes with more than one indicator, a sub-index was first calculated. The sub-indices were then
averaged to obtain the integrated watershed health index score (U.S. EPA, 2012). Depending on the specific
management objectives, it may be appropriate to place more weight on some ecological attributes than on others and
to use optional sub-indexes. At that point, the process becomes subjective and a logical decision framework can be
used to solicit and document expert opinion (Smith et al., 2003). Weighting was not used in this study for integrated
assessment. The normalized metrics, sub-index, and integrated watershed health index were calculated as follows:

$$Normalized\ metric\ value = \frac{Observed\ or\ simulated\ metric\ for\ watershed\ x}{Reference\ metric\ value\ for\ all\ watersheds\ in\ basin} \qquad (9)$$

$$Sub\text{-}index = \frac{(Normailzed\ metric\ 1 + Normalized\ metric\ 2 + \cdots + Normalized\ metric\ x)}{Total\ number\ of\ metrics} \qquad (10)$$

$$Watershed\ health\ index = \frac{(sub\text{-}index\ 1 + sub\text{-}index\ 2 + \cdots + sub\text{-}index\ x)}{Total\ number\ of\ sub\text{-}indices} \qquad (11)$$

<Table 4>

**3. Results and discussion**
3.1 Watershed health by each component in the Han River basin
Using the data reconstruction results for the six components of landscape, stream geomorphology, hydrology, water
quality, aquatic habitat condition, and biological condition, the watershed health analysis for each component was
conducted in 237 sub-watersheds as standard watersheds of the Han River basin. The sampling areas used to explain





the differences in watershed health results for each component were standard watersheds 101206 (urban 1.4% and
forest 88.1%), 100201 (urban 0.8% and forest 88.2%) and 101801 (urban 9.8% and forest 5%) (Figure 2a). The 101206,
100201, and 101801 standard watersheds are located in the upstream region of the Soyang Dam (SYD), in the
upstream region of the Chungju Dam (CJD), and in the downstream region of the Paldang Dam (PDD), respectively.
Figure 3 shows the landscape condition for green area (Figure 3a) and active river area (Figure 3b) indicators in
the Han River basin. Figure 12a shows the sub-index score for the watershed health assessment calculated according
to these two assessment indicators. The spatial patterns of watershed health for green areas were healthier for upstream
watersheds because the farther the watersheds are from the urban area, the greater in the increase in natural land cover.
The spatial patterns of watershed health for the active river area within 50 m of a stream were healthier for the upstream
watersheds for the same reason. For the 101206 standard watershed, the normalized values of the green area and the
active river area were 0.93 and 0.82, respectively, and the sub-index score of 0.89, which integrated the two normalized
values, indicated a very healthy watershed. For the 100201 standard watershed, the normalized values of the green
area and the active river area were 0.78 and 0.57, respectively, and the sub-index score of .0.66, which integrates the
two normalized values, indicates a less healthy watershed. In contrast, the 101801 standard watershed was revealed
to be in very poor health, with a score of 0.17 for the sub-index, while the normalized values of the green area and
active river area were 0.25 and 0.09, respectively. Hence, the study found that the downstream reaches of the Han
River basin are in greater need of green areas and active river areas compared to the upstream.
Figure 4 shows the stream geomorphology condition in the Han River basin. Figure 12b shows the sub-index score
for the watershed health assessment calculated using stream geomorphology indicators. The percentage of the length
of the assessed stream channel in reference condition was greater for the upstream watershed than the downstream
watershed. The high-gradient mountainous streams in the upstream watershed are characterized by relatively clean
streams that have not been subject to land cover modifications and river improvement work.
Figure 5 shows the SWAT model results for use in assessing the condition of hydrologic components PREC (a),
TQ (b), SQ (c), INFILT (d), SW (e), LQ (f), PERCOL (g), RECHARGE (h), and GWQ (i) for the period from 1985
to 2014 in the Han River basin. Figure 6 shows the SWAT model results for use in the water quality condition
assessment of the water quality constituents sediment (a), T-N (b), and T-P (c) for the same period in the Han River
basin. The sub-index results of the hydrologic and water quality conditions calculated are shown in Figure 12c and d,
respectively. The precipitation in the watershed directly affects the surface runoff and sediment transport and is the



most important factor impacting the maintenance of water quantity and can thus be used to identify areas critical for
maintaining watershed health. Nutrient (T-N and T-P) loads are often correlated with surface runoff and sediment
transport rates (USDA-SCS, 1972). The fugitive sediment from the landscape is carried by overland flow (surface
runoff), and the dominant pathway for nitrate loss is through leaching to groundwater and then via baseflow (Randall
and Mulla, 2001).
The sub-indices of hydrologic condition calculated by the four hydrologic classifications, such as the total metric
(PREC and TQ), surface processes metric (SQ), soil water dynamics metric (INFILT, SW, and LQ), and groundwater
dynamics metric (PERCOL, RECHARGE, and GWQ), and the water quality condition calculated by sediment, T-N,
and T-P were split into three periods of ten years—1985–1994, 1995–2004, and 2005–2014—for the assessment of
changes over time (Figure 9). The test areas used to explain the differences in the results of watershed health the for
hydrologic and water quality components are the SYD watershed and CJD watershed located in the upstream region
and the PDD watershed and lower watershed located in the downstream region (Figure 2c). For the SYD watershed
(Figure 9a), the watershed health scores of the surface water, soil water, and groundwater hydrology increased in the
recent past compared to the period 1985–1994 due to the slight increases in PREC and TQ; thus, the watershed water
quality was diminished. The health of the hydrology in the CJD watershed showed a decreased tendency in contrast
to the SYD watershed as a result of the decrease in PREC and TQ (Figure 9b). In the case of the PDD watershed and
the lower watershed, the groundwater of the PDD watershed was not sufficient, but overall watershed health scores
remained within their reference levels (approximately 0.5) (Figure 9c and d). This water quantity stress (large volume
of water in the stream) may have negative effects on water quality, with a decreased watershed health score for the
sediment, T-N, and T-P. In particular, the SYD watershed was rich in soil water and the CJD watershed was rich in
surface and groundwater.
Figure 10 shows the watershed health index score changes for the hydrologic and water quality conditions during
1995–2004 and the most recent ten years (2005–2014) based on the reference period (1985–1994). Improved health,
deteriorating health, and no change area in the Han River basin are illustrated with green, red, and white, respectively.
On the whole, the watershed hydrologic condition was better in the North Han River basin compared to the South Han
River basin. In particular, during the last ten years (Figure 10b), the watershed health was poorer due to worse results
for the surface processes metric and soil water dynamics compared to the 1995–2004 period (Figure 10a). However,
in the case of water quality, during the last ten years (Figure 10d), the watershed health increasingly improved in parts





of Han River basin compared to 1995–2004 (Figure 10c), while the water quality of the Chungju dam (CJD) watershed
was growing worse. The water quality policy of South Korea, developed after years of hard work and high costs, thus
resulted in some improvements.
Figure 11 shows the overlay results (Figure 11c) showing the poor watershed health of both hydrology (Figure 11a)
and water quality (Figure 11b). The five poor levels of hydrology and water quality were calculated as the difference
between (b) and (a) of Figure 10 and between (d) and (c) of Figure 10, respectively. The spatial distributions of poor
watershed health levels allow us to understand the vulnerable areas in parts of the CJD watershed, the upstream SYD
watershed, and the downstream PDD watershed with respect to hydrology and water quality.

<Figure 9>
<Figure 10>
<Figure 11>

Figure 7 shows the aquatic habitat condition for the aquatic habitat connectivity (Figure 7a) and wetland (Figure
7b) indicators in the Han River basin. Figure 12e shows the sub-index score for the watershed health assessment
calculated according to these two assessment indicators. The spatial distribution patterns of the reservoirs for aquatic
habitat connectivity were concentrated in the downstream areas of the Han River basin. The spatial distribution
patterns of the wetlands seem to follow a similar pattern. For the 101206 standard watershed, the normalized values
of the aquatic habitat connectivity and wetland were 0.00 (no reservoir) and 0.99, respectively, and the sub-index score
of 0.90, which integrates the two normalized values, indicates a very healthy watershed. In contrast, for the 100201
standard watershed, the normalized values of the aquatic habitat connectivity and wetland were 0.46 and 0.34,
respectively, and the sub-index score of 0.28, which integrates the two normalized values, indicated an unhealthy
watershed. At the 101801 standard watershed, the aquatic habitat condition results from the aquatic habitat
connectivity (0.77) and wetland (0.66) indicators show a relatively high value of 0.68.
The biological pollution classes of the TDI, BMI, and FAI were examined by ecoregion and river basin (Figure 8).
These relationships were found to be significantly correlated. In the downstream areas, the TDI, BMI, and FAI are
worse. However, the degree to which the TDI, BMI and FAI predict trophic diatom, benthic macroinvertebrate, and
fish communities depends on the presence and levels of other stressors, such as large amounts of chlorophyll-a (Chl-





a), low dissolved oxygen (DO) and biochemical oxygen (BOD), and high temperature. The normalized values of TDI,
BMI and FAI were 0.70, 0.98, and 0.92, respectively, in the 101206 standard watershed located upstream; 0.69, 0.98,
and 0.72, respectively, in the 100201 standard watershed located upstream; and 0.32, 0.25, and 0.25, respectively, in
the 101801 standard watershed located downstream.. The sub-index analysis of the TDI, BMI, and FAI was completed
except in the no-data areas (North Korea) in the Han River Basin (Figure 12f). The sub-index scores integrating the
three normalized values were 0.91 and 0.83 for the 101206 and 100201 standard watersheds, respectively, indicating
very healthy watersheds, and the sub-index score of 0.26 at the 101801 standard watershed indicated an unhealthy
watershed.
The outputs of the watershed health provide basic data for local communities to proactively plan for growth. The
sub-index results of the watershed health assessment for each component can be optionally used to guide the master
planning process for watershed management at the watershed scale depending on the specific management objectives
and can be combined with any of the other sub-indices in the Han River basin for use in determining priority
conservation areas.

3.2 Assessment of integrated watershed health
To assess the overall watershed health in the Han River basin, the results of the individual assessments were
synthesized to provide an integrated watershed health index score for the thirty-year period (1985–2014). The sample
areas used to explain the differences in watershed health results for each component were standard watersheds 101206
(urban 1.4% and forest 88.1%), 100201, (urban 0.8% and forest 88.2%) and 101801 (urban 9.8% and forest 55.7%)
(Figure 2a). The 101206, 100201, and 101801 standard watersheds were located in the upstream region of the Soyang
dam (SYD), in the upstream region of the Chungju dam (CJD), and in the downstream region of the Paldang dam
(PDD), respectively.
Figure 12 displays the normalized scores for each of the six attribute sub-indices and integrated watershed health score.
The integrated watershed health exhibited a decreased tendency farther down the watershed. The integrated watershed
health of the 101206 and 100201 standard watersheds was revealed to be very good, with ratings of 1 and 0.91,
respectively. However, the 101206 standard watershed exhibited distinctive weakness with respect to hydrologic
condition (0.06), especially in the surface (0.16) and groundwater (0.17). Although the 100201 standard watershed
was a very healthy watershed, like the 101206 watershed, it showed a distinctive weakness with respect to water





quality (0.1) and aquatic habitat condition (0.28). It is important to develop systematic plans to suit watershed
circumstances and characteristics so that watershed management is more effective. The 101801 watershed was
revealed to be in poor health, with a water quality rating of 0.25. This area requires urgent action to restore the
landscape, water quality, and biological conditions and to protect the water quantity. Table 5 shows watershed health
scores in test areas (upper/lower stream) of the Han River basin.

<Figure 12>
<Table 5>

**4. Conclusions**
In this study, a watershed health assessment of the Han River basin in South Korea was performed using monitoring
data and SWAT modeling results. Six essential indicators of healthy watersheds were used in the assessment: 1)
landscape condition, 2) geomorphology, 3) hydrology, 4) water quality, 5) habitat, and 6) biological condition. In
particular, the sub-index of watershed health related to hydrology and water quality was developed to assess the
possible long-term changes in the watershed using SWAT modeling results.
During the most recent ten-year period (2005–2014), the watershed health declined, as indicated by the worse
results for the surface processes metric and soil water dynamics compared to the 1995–2004 period. The spatial
distributions of poor watershed health levels revealed the vulnerable areas in parts of the CJD watershed, upstream of
the SYD watershed, and downstream of the PDD watershed with respect to hydrology and water quality.
The sub-index results of the watershed health assessment for each component can be used to guide the master
planning process for watershed management at the watershed scale based on specific management objectives and can
be combined with any of the other sub-indices in the Han River basin for use in determining priority conservation
areas.

**Acknowledgments**
This research was supported by a grant (14AWMP-B082564-01) from the Advanced Water Management Research
Program funded by the Ministry of Land, Infrastructure and Transport of the Korean government.





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




Figure 1. Flowchart of the study procedure for the watershed health assessment.

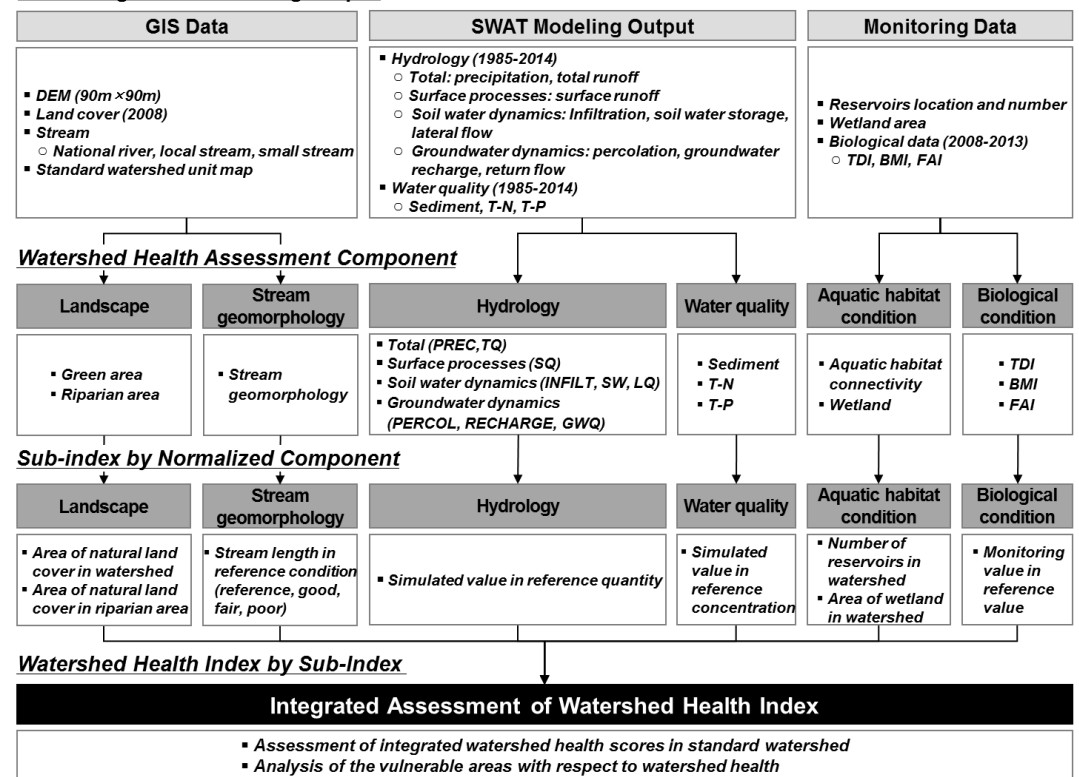





Figure 2. Locations of (a) the Han River basin boundaries and gauging stations for the watershed (SWAT) modeling,
(b) land cover classification, and (c) test area.

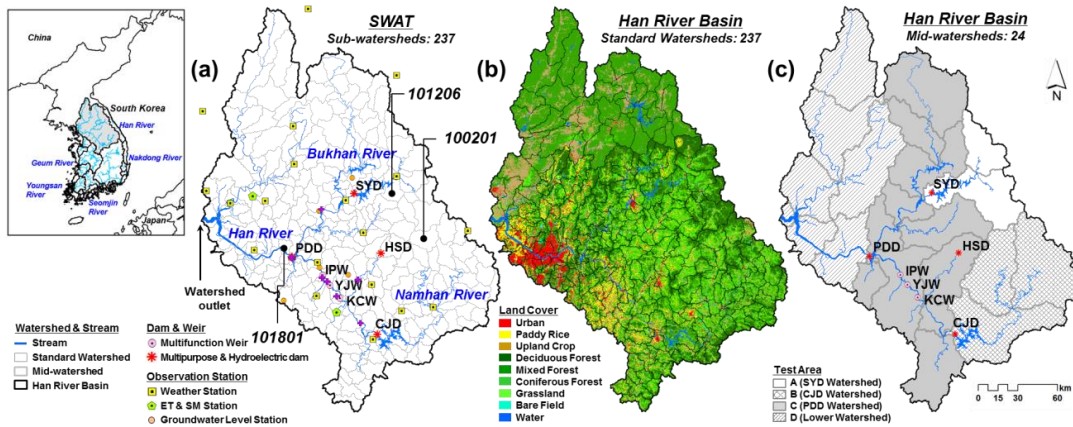






Figure 3. Landscape condition for (a) green area and (b) riparian area.

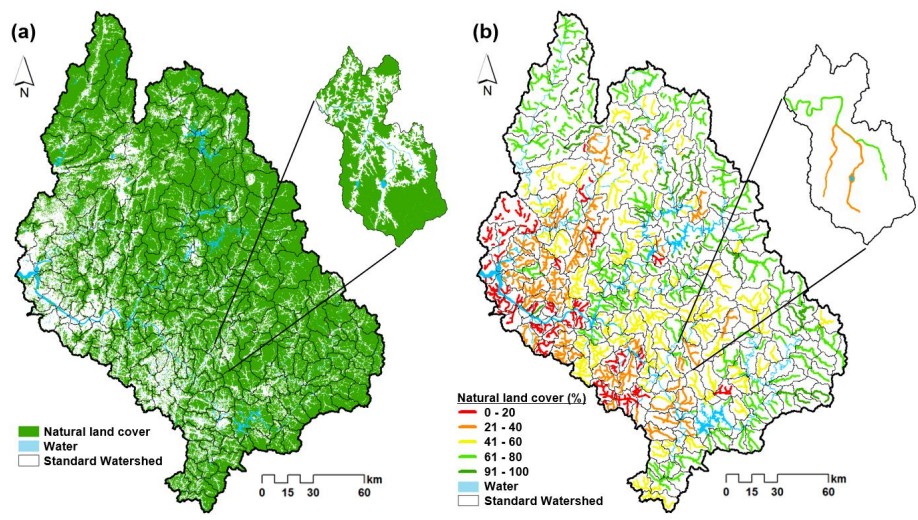




Figure 4. Stream geomorphic conditions: (a) stream order and (b) stream geomorphic conditions.

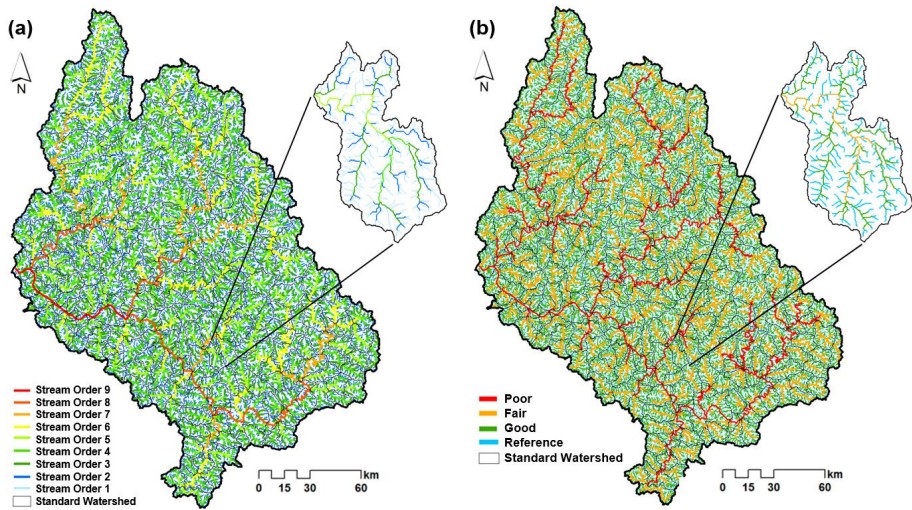





Figure 5. Hydrologic condition for (a) precipitation, (b) total runoff, (c) surface runoff, (d) infiltration, (e) soil water
storage, (f) lateral flow, (g) percolation, (h) groundwater recharge, and (b) return flow according to the hydrological
(SWAT) modeling for the period from 1985 to 2014 in the Han River basin.





Figure 6. Water quality condition for (a) sediment, (b) T-N and (c) T-P according to the hydrological (SWAT) modeling
for the period from 1985 to 2014 in the Han River basin.

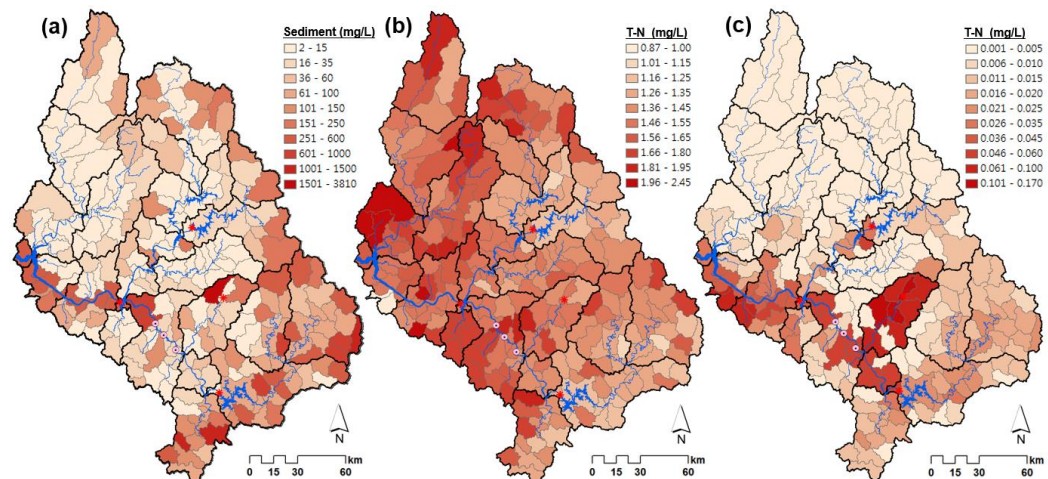




Figure 7. Aquatic habitat conditions for (a) aquatic habitat connectivity and (b) wetland.

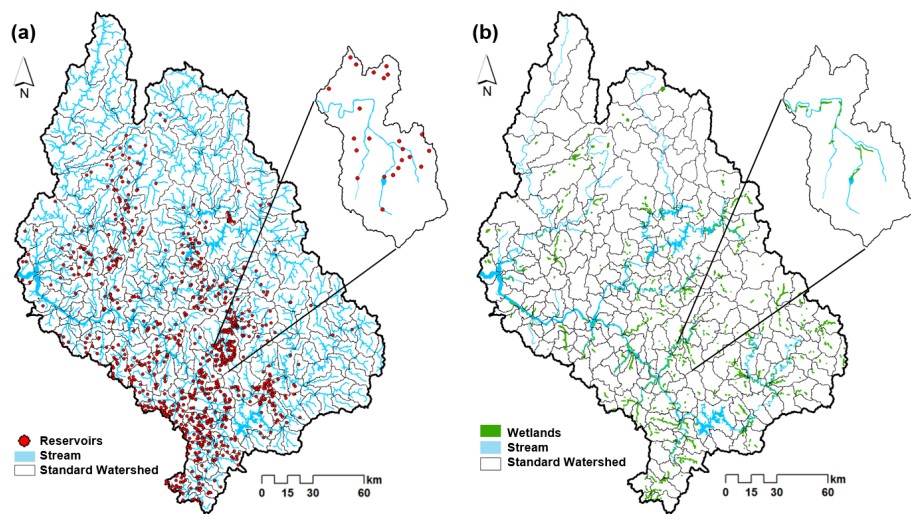





Figure 8. Biological conditions of (a) FAI, (b) BMI and (c) FAI according to the observed monitoring data for the
period from 2008 to 2013 in the Han River basin.

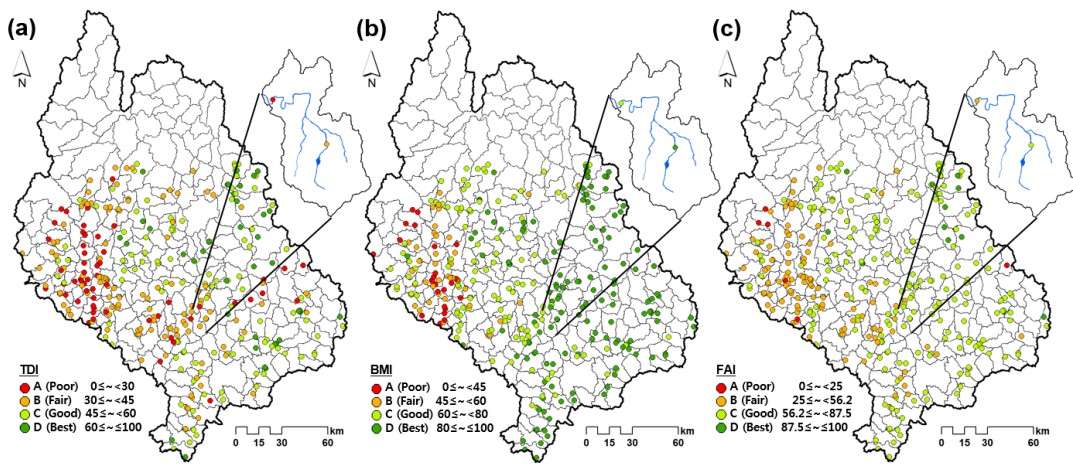






Figure 9. Change in hydrology and water quality for (a) A (SYD watershed), (b) B (CJD watershed), (c) C (PDD
watershed), and (d) D (lower watershed) test areas for three ten-year periods.

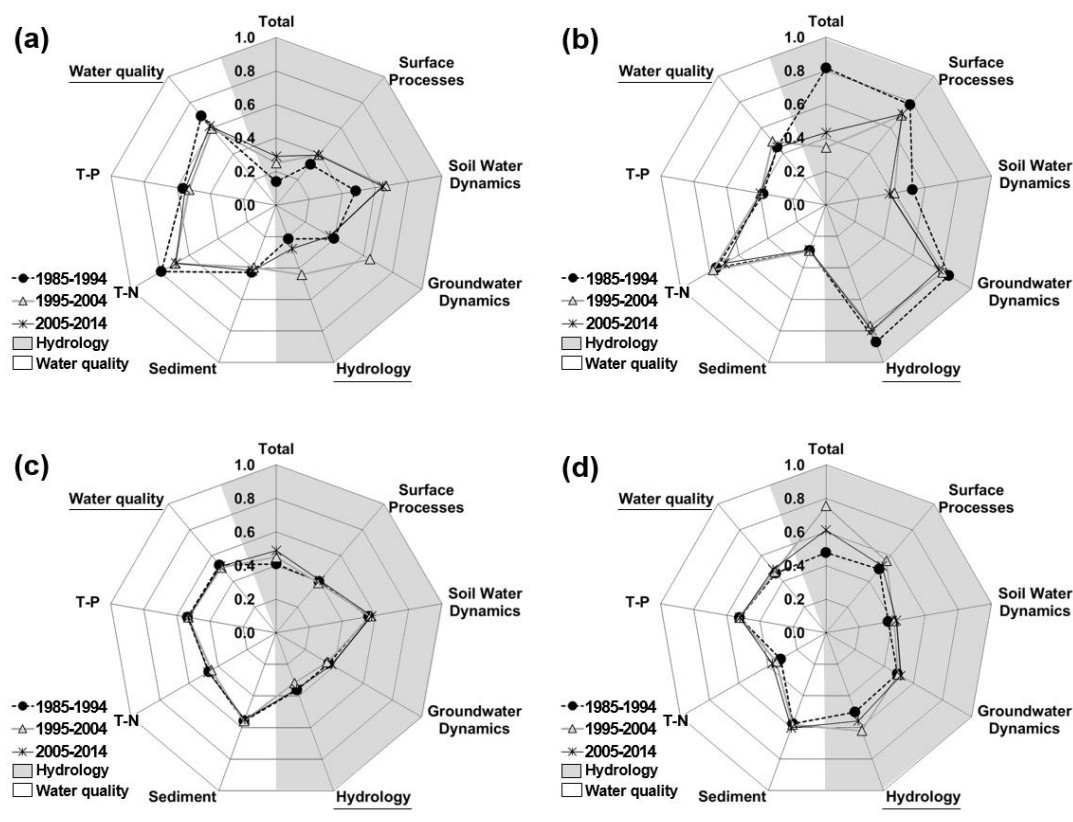






Figure 10. The watershed health index score changes for the hydrologic (a and b) and water quality (c and d) conditions
during the period 1995–2004 and the most recent ten-year period (2005–2014) based on the reference period (1985–

710    1994).

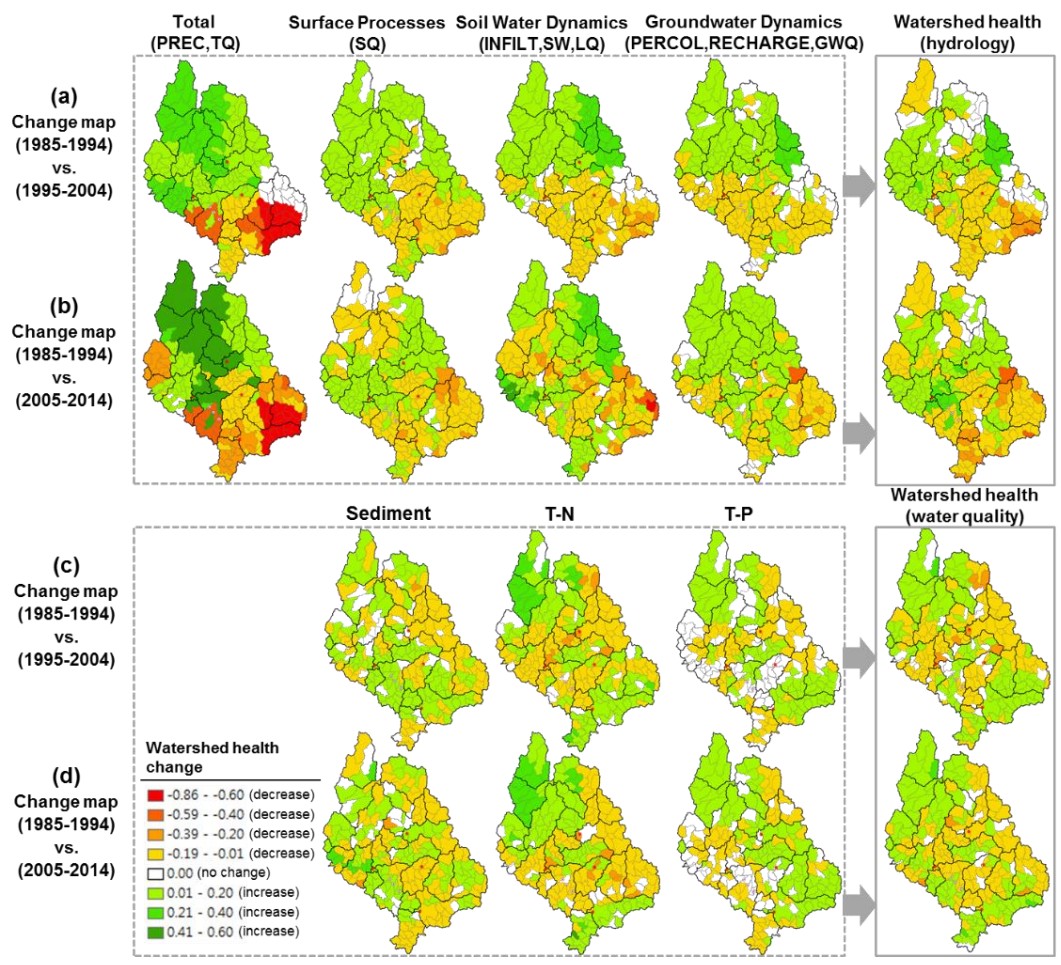




Figure 11. The poor watershed health revealed by (a) hydrology, (b) water quality, and (c) overlay results.

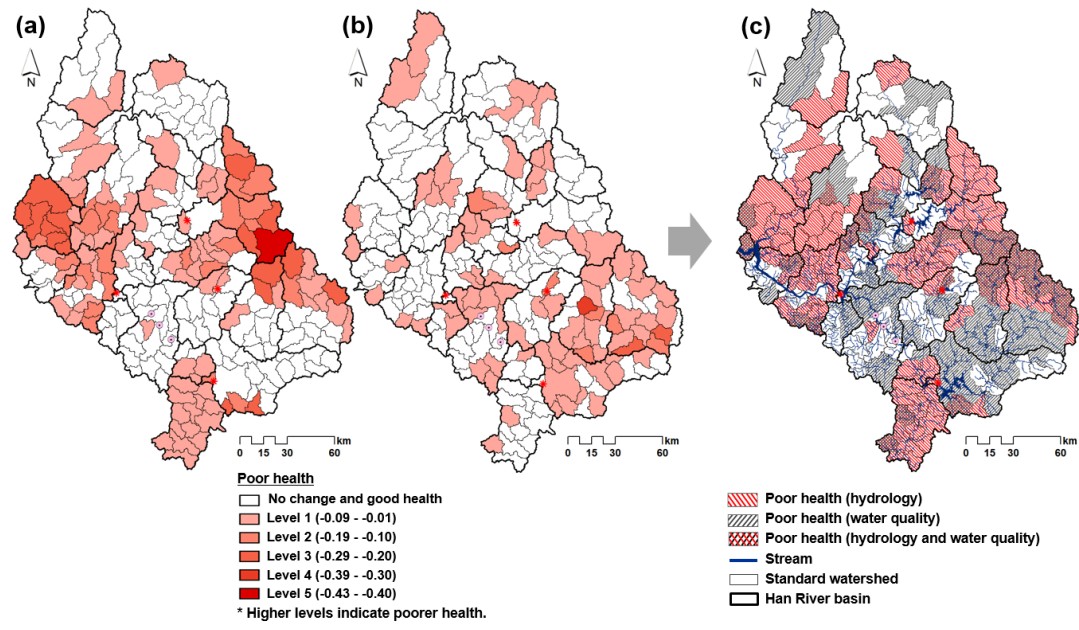






Figure 12. The results of the watershed health index for (a) landscape, (b) stream geomorphology, (c) hydrology, (d)
water quality, (e) aquatic habitat, (f) biological condition, and (g) integrated watershed health.

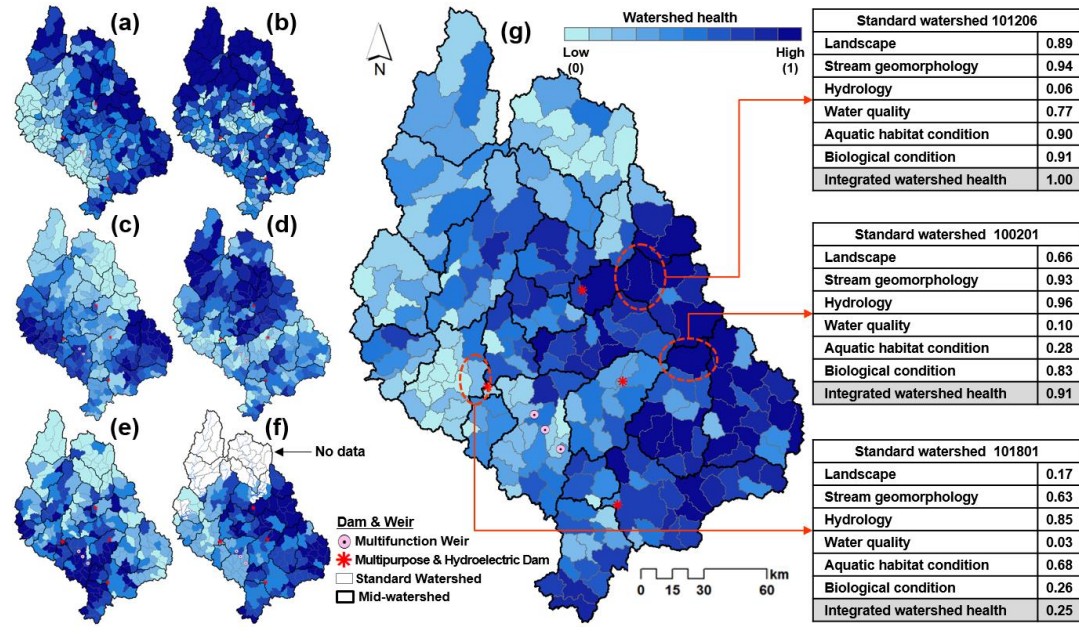






Table 1 Metrics and summary dataset used for the assessment of watershed health in the study watershed

| Component (metric) | Measurement method | Dataset |
|---|---|---|
| *Landscape* | | *GIS data* |
| Green infrastructure metric | Percentage of watershed occupied by natural land cover | Land cover 2008[a] |
| Active river area metric | Percentage of natural land cover within the active river area | Land cover 2008, stream[b] |
| *Geomorphology* | | *GIS data* |
| Stream geomorphology metric | Percentage of assessed stream length in reference condition | SRTM DEM (90×90)[c], stream |
| *Hydrology* | | *SWAT modeling data (1985–2014)* |
| Total metric | Precipitation and total runoff storage ratio | PREC, TQ |
| Surface processes metric | Surface runoff storage ratio | SQ |
| Soil water dynamics metric | Infiltration, soil water and lateral flow storage ratio | INFILT, SW, LQ |
| Groundwater dynamics metric | Percolation, groundwater recharge and return flow storage ratio | PERCOL, RECHARGE, GWQ |
| *Water quality* | | *SWAT modeling data (1985–2014)* |
| Water quality metric | Percentage of assessed value in reference criteria | Sediment, T-N, T-P |
| *Aquatic habitat condition* | | *GIS data* |
| Habitat connectivity metric | Reservoir density (number of reservoirs per stream length) | Reservoir location map[d], stream |
| Wetland metric | Percentage of watershed occupied by wetlands | Land cover 2008 |
| *Biological condition* | | *Monitoring data (2008–2013)[e]* |
| Biological metric | Percentage of assessed score in reference condition | TDI, BMI, FAI |

Main data sources included [a] the Korea Ministry of Environment (KME); [b] the Ministry of Land, Infrastructure, and Transport (MOLIT) in
South Korea; [c] the International Center for Tropical Agriculture (CIAT); [d] the Korea Rural Community Corporation (KRC); and [e] the Korea
Ministry of Environment (KME) in South Korea (Ministry of Environment, 2013).





Table 2 Calibration and validation results for dam inflow, dam storage volume, evapotranspiration and soil moisture,
groundwater level fluctuation, sediments, T-N, and T-P at each calibration point.

| Model output | Evaluation criteria | Cal. | Val. | Cal. | Val. | Cal. | Val. | Cal. | Val. | Cal. | Val. | Cal. | Val. | Cal. | Val. |
|---|---|---|---|---|---|---|---|---|---|---|---|---|---|---|---|
| Dam inflow (mm) | Locations | HSD | | SYD | | CJD | | KCW | | YJW | | IPW | | PDD | |
| | $R^2$ | 0.82 | 0.84 | 0.90 | 0.89 | 0.81 | 0.74 | 0.90 | 0.63 | 0.91 | 0.62 | 0.93 | 0.59 | 0.92 | 0.88 |
| | NSE | 0.61 | 0.57 | 0.78 | 0.78 | 0.63 | 0.58 | 0.78 | 0.79 | 0.77 | 0.76 | 0.81 | 0.95 | 0.83 | 0.76 |
| | RMSE (mm/day) | 7.9 | 9.3 | 3.8 | 3.9 | 3.5 | 3.1 | 6.5 | 0.7 | 9.1 | 2.4 | 9.2 | 2.9 | 0.8 | 2.3 |
| | PBIAS (%) | 14.5 | 12.5 | 10.3 | 14.0 | 8.9 | 9.9 | 18.0 | 4.9 | 25.5 | 14.1 | 25.6 | 17.2 | 2.2 | 6.8 |
| Dam storage ($10^6$ m$^3$) | | HSD | | SYD | | CJD | | KCW | | YJW | | IPW | | PDD | |
| | $R^2$ | 0.73 | 0.77 | 0.94 | 0.96 | 0.87 | 0.84 | 0.57 | 0.85 | 0.47 | 0.83 | 0.47 | 0.79 | 0.40 | 0.44 |
| | PBIAS (%) | 18.9 | 9.9 | 16.3 | 9.3 | 18.2 | 15.2 | 5.1 | 7.4 | 3.7 | 11.1 | 9.1 | 7.2 | 0.9 | 1.4 |
| Evapotrans-piration (mm) | Locations | SM | | CM | | - | | - | | - | | - | | - | |
| | $R^2$ | 0.81 | 0.73 | 0.70 | 0.74 | - | - | - | - | - | - | - | - | - | - |
| | NSE | 0.64 | 0.45 | 0.50 | 0.55 | - | - | - | - | - | - | - | - | - | - |
| | RMSE (mm/day) | 2.3 | 9.1 | 4.0 | 3.0 | - | - | - | - | - | - | - | - | - | - |
| | PBIAS (%) | 9.6 | 30.2 | 11.6 | 23.7 | - | - | - | - | - | - | - | - | - | - |
| Soil moisture (%) | Locations | SM | | CM | | - | | - | | - | | - | | - | |
| | $R^2$ | 0.85 | 0.75 | 0.78 | 0.78 | - | - | - | - | - | - | - | - | - | - |
| Grounwater level (EL.m) | Locations | - | | - | | GPGP | | YPGG | | YPYD | | YIMP | | HCGD | |
| | $R^2$ | - | - | - | - | 0.70 | 0.63 | 0.64 | 0.45 | 0.70 | 0.41 | 0.53 | 0.40 | 0.69 | 0.67 |
| Sediment (ton) | Locations | SG | | CSG | | JW | | KCW | | YJW | | IPW | | PDD | |
| | $R^2$ | 0.78 | 0.70 | 0.78 | 0.76 | 0.90 | 0.71 | 0.54 | 0.64 | 0.84 | 0.54 | 0.69 | 0.66 | 0.72 | 0.80 |
| T-N (kg) | $R^2$ | 0.58 | 0.71 | 0.64 | 0.71 | 0.82 | 0.68 | 0.50 | 0.61 | 0.52 | 0.49 | 0.46 | 0.62 | 0.66 | 0.62 |
| T-P (kg) | $R^2$ | 0.77 | 0.77 | 0.88 | 0.88 | 0.80 | 0.56 | 0.56 | 0.58 | 0.50 | 0.47 | 0.66 | 0.70 | 0.74 | 0.69 |

[a] Cal. = calibration period (HSD, SYD, CJD and PDD: 2005-2009, KCW, YJW and IPW: 2013) and Val. = validation period (HSD,
SYD, CJD and PDD: 2010-2014, KCW, YJW and IPW: 2014)






Table 3 Description of the stream geomorphic condition categories (Kline et al., 2009) and stream order for watershed
health assessment of geomorphic condition in the Han River basin

| Condition | Description | River classification | Stream order (1–9) |
|---|---|---|---|
| Reference | In Equilibrium – no apparent or significant channel, floodplain, or land cover modifications; channel geometry is likely to be in balance with the flow and sediment produced in its watershed. | Mountainous river | 1 |
| Good | In Equilibrium but may be in transition into or out of the range of natural variability – minor erosion or lateral adjustment but adequate floodplain function; any adjustment from historical modifications nearly complete. | Small river | 2–3 |
| Fair | In Adjustment – moderate loss of floodplain function or moderate to major plan-form adjustments that could lead to channel avulsions. | Local river | 4–5 |
| Poor | In Adjustment and Stream Type Departure – may have changed to a new stream type, or central tendency of fluvial processes or significant channel and floodplain modifications may have altered the channel geometry such that the stream is not in balance with the flow and sediment produced in its watershed. | Urban river, National river | 6–9 |




Table 4 Summary of hydrology, water quality and biological criteria used to screen for reference condition in the Han
River basin

| Component | Source | Reference condition |
|---|---|---|
| *Hydrology* | | |
| Precipitation | River basin average of 30 years (1985–2014) simulated by SWAT | 1,395.1 (mm) |
| Total runoff | | 919.5 (mm) |
| Surface runoff | | 249.4 (mm) |
| Infiltration | | 726.4 (mm) |
| Soil water storage | | 85.3 (mm) |
| Lateral flow | | 345.9 (mm) |
| Percolation | | 363.8 (mm) |
| Groundwater recharge | | 22.9 (mm) |
| Return flow | | 324.2 (mm) |
| *Water quality* | | |
| Sediment | The levels greater than "marginally good" level on a seven-point scale | 15 (mg/L) |
| T-N | (excellent, very good, good, marginally good, fair, poor, very poor) of water | 0.6 (mg/L) |
| T-P | quality criteria for streams and lakes devised by the Basic Environmental Policy Act (BEPA) in South Korea. | 0.05 (mg/L) |
| *Biological condition* | | |
| TDI | The "best" and "good" levels on a four-point scale (best, good, fair and poor) | 72.5 |
| BMI | of biological condition criteria devised by the Korea Ministry of | 80.0 |
| FAI | Environment (KME) (Ministry of Environment, 2013). | 78.1 |






Table 5 Results of watershed health score in each test area (upper/lower stream) of the Han River basin

| Component | A (SYD watershed) | B (CJD watershed) | C (PDD watershed) | D (Lower watershed) |
|---|---|---|---|---|
| *Landscape* | **0.80** | **0.66** | **0.53** | **0.26** |
| Green infrastructure metric | 0.85 | 0.67 | 0.52 | 0.25 |
| Active river area metric | 0.74 | 0.65 | 0.53 | 0.28 |
| *Geomorphology* | **0.75** | **0.47** | **0.46** | **0.54** |
| *Hydrology* | **0.21** | **0.74** | **0.37** | **0.60** |
| Total | 0.19 | 0.51 | 0.44 | 0.65 |
| Surface processes | 0.36 | 0.73 | 0.40 | 0.53 |
| Soil water dynamics | 0.61 | 0.44 | 0.58 | 0.39 |
| Groundwater dynamics | 0.30 | 0.55 | 0.45 | 0.58 |
| *Water quality* | **0.63** | **0.45** | **0.52** | **0.48** |
| Sediment | 0.40 | 0.29 | 0.55 | 0.61 |
| T-N | 0.76 | 0.70 | 0.49 | 0.32 |
| T-P | 0.52 | 0.40 | 0.53 | 0.53 |
| *Aquatic habitat condition* | **0.39** | **0.43** | **0.55** | **0.45** |
| Habitat connectivity | 0.22 | 0.30 | 0.52 | 0.40 |
| Wetland | 0.53 | 0.51 | 0.49 | 0.41 |
| *Biological condition* | **0.92** | **0.73** | **0.47** | **0.23** |
| TDI | 0.83 | 0.67 | 0.50 | 0.25 |
| BMI | 0.88 | 0.78 | 0.46 | 0.22 |
| FAI | 0.92 | 0.70 | 0.47 | 0.27 |
| *Integrated assessment* | **0.82** | **0.75** | **0.47** | **0.30** |
