# Peer review of "Assessment of Integrated Watershed Health based on Natural Environment,"

_Hydrology and Earth System Sciences, 2017_

## Referee Comment (RC1) · Anonymous Referee #1 · 25 Mar 2017

This study evaluated health condition of a watershed of the Han River basin (34,148 km$^2$) in South Korea was performed using monitoring data and SWAT modeling results. Six essential indicators of healthy watersheds were used in the assessment: landscape condition, geomorphology, hydrology, water quality, habitat, and biological condition. The research findings from this study provided guidance for watershed management at the watershed scale based on specific management objectives and can combined with any of the other sub-indices in the Han River basin for use in determining priority conservation areas. This paper is well organized and well written generally. Detailed method description was incorporated. The scientific results and conclusions

were presented in a clear, concise, and well-structured way. The number and quality of references is appropriate. But method and results should be reduced. The importance of six essential indicators of healthy watersheds was not well described. More in-depth discussion should be included to support the interpretations and conclusions. This manuscript can be reviewed after major revisions. What is the novel idea this manuscript provided to scientific knowledge? Please describe it and use your results and discussion to support it.

The last sentence of the abstract "The results suggest that approaches aimed at simultaneously improving the water quality, hydrology, and aquatic ecology conditions may be necessary to improve integrated watershed health." Is this the scientific questions being answered in this manuscript? Please provide specific discussion of results and summarize them in conclusion to support this point. Otherwise, I do not think this sentence should be here.

2.4 Hydrology and water quality simulations using the SWAT model: the session is mainly focus on basic information about SWAT. If it is not specific for your project, it is better to put information in the Introduction rather than in Methods. And authors already described data collection related to SWAT model setup and SWAT outputs in 2.3, thus it is better to introduce SWAT model before discussing data related to it.

Is 90 m grid size DEM data sufficient to accurately simulate hydrology and water quality at such a large area? Is there any higher resolution elevation data can be used?

Is calibration period (2005–2009) and validation period (2010–2014) both incorporate wet and dry years?

Statistical evaluation criteria R2, NSE and PBIAS are all sensitive to high values. Criteria less sensitive to high values, such as Modified NSE and KGE, may could be incorporated.

Page 8 line 197: this paragraph described a lot of detailed information about dams. It

is better to condense it and save more space for in-depth discussions. How was dam information being set in SWAT model?

Page 9 line 226: "The calibrated parameters and hydrograph of the calibration results in the Han River basin were described by 227 Chung et al (2017)." Parameter definition, physical meaning, range used for calibration and calibrated values are very important information. Please describe this information in supplementary materials to prove that your calibration and validation is reliable.

Results and discussion generally is redundant. This part need to be condensed. Some information can be incorporated in supplementary materials.

Page 10 line 237: "T-N was between 0.46 and" There should be a space between "0.46" and "and".

Page 10 line 239: should there have a space before and after $\geq$ ?

Page 19 line 478: Please improve wording of the first sentence.

Conclusion did not interpolate researching findings well. The results showed the watershed health declined and targeted the vulnerable areas, but what is boarder impacts of these results? How will it be beneficial for watershed management? It would be more meaningful if authors can incorporate this information.

What is limitation of this study, such as water quantity, quality data, or model input limitations? How to improve it in the further study? What kind of take-home messages you would like to delivery to readers?

---

## Referee Comment (RC2) · Anonymous Referee #2 · 20 Apr 2017

This study assesses health condition of the Han River basin in South Korea based on monitoring data, water quantity and quality time series simulations of the SWAT model and an ensemble of indicators related to 6 components of the watershed landscape related to stream geomorphology, hydrology, water quality, aquatic habitat condition, and biological condition. The paper deals with an interesting topic which is watershed health condition. Indeed, there is a weak understanding of the complex processes and watershed components interactions that govern the healthy/unhealthy state of the watershed and such paper is needed to bridge the gap. This is a nice paper, well written and structured in a coherent way. But to my opinion, the approach needs to be

improved by including an uncertainty assessment/analysis of the SWAT model.

Authors used SWAT model simulations for water quality and quantity time series reconstruction which in-turn were used for indicators and sub-index development, as stated in the first specific object of the paper. Rely on model simulation for developing these indicators may add uncertainty in the indicators and sub-indexes. In addition, the definition of the reference condition here is crucial and used as a kind of "threshold" to discriminate between healthy and unhealthy watershed condition. This choice is based on SWAT simulation without any uncertainty analysis. I would prefer to see an acceptable range of reference condition based on model uncertainty analysis rather a single value of reference indicator.

L.314-316. Authors mentioned that surface water and lateral groundwater flow interactions were of major importance for the water balance in the Han River basin. In particular, infiltration, return flow, groundwater recharge were important factors for the whole hydrological cycle. These results were based on SWAT simulations. Again, in absence of model uncertainty analysis the contribution of these components to the total water balance may vary or change depending on the parameter of the model. Therefore, I don't think that metrics developed based on the above results can be used for establishing specific management objectives as stated by the authors in line 323.

---

## Author Comment (AC1) · 25 May 2017

MS No.: hess-2017-88 MS Type: Research article Special Issue: Coupled terrestrial-aquatic approaches to watershed-scale water resource sustainability Title: Title: Assessment of Integrated Watershed Health based on Natural Environment, Hydrology, Water Quality, and Aquatic Ecology Journal: Hydrology and Earth System Sciences

Anonymous Referee #1

COMMENTS: This study evaluated health condition of a watershed of the Han River basin (34,148 km2) in South Korea was performed using monitoring data and SWAT
modeling results. Six essential indicators of healthy watersheds were used in the assessment: landscape condition, geomorphology, hydrology, water quality, habitat, and biological condition. The research findings from this study provided guidance for watershed management at the watershed scale based on specific management objectives and can combined with any of the other sub-indices in the Han River basin for use in determining priority conservation areas. This paper is well organized and well written generally. Detailed method description was incorporated. The scientific results and conclusions were presented in a clear, concise, and well-structured way. The number and quality of references is appropriate. But method and results should be reduced. The importance of six essential indicators of healthy watersheds was not well described. More in-depth discussion should be included to support the interpretations and conclusions. This manuscript can be reviewed after major revisions. What is the novel idea this manuscript provided to scientific knowledge? Please describe it and use your results and discussion to support it.

1. The last sentence of the abstract "The results suggest that approaches aimed at simultaneously improving the water quality, hydrology, and aquatic ecology conditions may be necessary to improve integrated watershed health." Is this the scientific questions being answered in this manuscript? Please provide specific discussion of results and summarize them in conclusion to support this point. Otherwise, I do not think this sentence should be here.

\* Response: (Lines 25-27) We removed the last sentence of the abstract. And we revised this as follows: "As a result, during the most recent ten-year period of 2005–2014, the watershed health declined, as indicated by the worse results for the surface processes metric and soil water dynamics compared to the 1995–2004 period. The integrated watershed health tended to decrease farther downstream the watershed."

2. 2.4 Hydrology and water quality simulations using the SWAT model: the session is mainly focus on basic information about SWAT. If it is not specific for your project, it is better to put information in the Introduction rather than in Methods. And authors

already described data collection related to SWAT model setup and SWAT outputs in 2.3, thus it is better to introduce SWAT model before discussing data related to it.

* Response: (Lines 148-157, and 193-200) 2.4 Hydrology and water quality simulations using the SWAT model: the session is mainly focus on not only basic information about SWAT but also model calibration and validation for hydrology and water quality simulation data. The information of this session are very important as methods for watershed health assessment. We added a new session 2.3 SWAT model description before 2.4 Data collection and removed basic information about SWAT in 2.5 Hydrology and water quality simulations using the SWAT model.

3. Is 90 m grid size DEM data sufficient to accurately simulate hydrology and water quality at such a large area? Is there any higher resolution elevation data can be used?

* Response: (Lines 206-207) Our study area included parts of North Korea. We have 30 m DEM covered by South Korea, but we don't have data in North Korea. Therefore, we used a world 90 m DEM from the Shuttle Radar Topography Mission (SRTM) of the International Centre for Tropical Agriculture (CIAT). As shown in the below figures, the results for hydrology and water quality were reasonable. I think that precipitation has an even greater impact on the hydrologic simulations than the DEM resolution does. In addition, the resolution of 90 m DEM deems appropriate for simulating the watershed hydrology for the 237 sub-watersheds (average area is 144 km$^2$) using the SWAT model.

4. Is calibration period (2005–2009) and validation period (2010–2014) both incorporate wet and dry years?

* Response: (Lines 241-246) We incorporated both wet and dry years in calibration period (2005–2009) and validation period (2010–2014). The average annual precipitation of Han River basin is 1,300 mm. For the calibration period (2005–2009), wet and dry years are 2006 (1,625 mm) and 2008 (1,160 mm). For the validation period

(2010–2014), wet and dry years are 2011 (1,640 mm) and 2014 (734 mm).

5. Statistical evaluation criteria R2, NSE and PBIAS are all sensitive to high values. Criteria less sensitive to high values, such as Modified NSE and KGE, may could be incorporated.

* Response: (Lines 281-284) We added NSE with inverse discharge (1/Q) in Table 2. We added new sentences: "Additionally, model calibration and validation included the NSE with inverse discharge (1/Q) for low flow. The average NSE with inverse discharge (1/Q) during the calibration (2005–2009) and validation (2010–2014) periods was 0.35 at HSD, 0.53 at SYD, 0.30 at CJD, 0.54 at KCW, 0.47 at YJW, 0.69 at IPW, and 0.58 at PDD."

6. Page 8 line 197: this paragraph described a lot of detailed information about dams. It is better to condense it and save more space for in-depth discussions. How was dam information being set in SWAT model?

* Response: (Lines 211-219 and 223-235) We removed paragraph that is about description of detailed dam informations. We addedd new sentences about dam information being set in SWAT model as follows: "The flow and water quality of the Han River are impacted by the discharge operations of these large dams and weirs; therefore, dam and weir operations must be incorporated into the modeling framework to enable successful modeling. In the SWAT model, dam operations are modeled based on measured daily discharges, measured monthly discharges, average annual discharges, or target storage volumes. In this study, the measured daily discharges from the four dams and three weirs were directly imported into the SWAT model."

7. Page 9 line 226: "The calibrated parameters and hydrograph of the calibration results in the Han River basin were described by 227 Chung et al (2017)." Parameter definition, physical meaning, range used for calibration and calibrated values are very important information. Please describe this information in supplementary materials to prove that your calibration and validation is reliable.

* Response: (Lines 260-268) We added new sentences about information for parameter definition and physical meaning as follows: "In this study, both calibration and validation were manually performed using a trial-and-error approach within recommended ranges to maximize the expert knowledge of watershed characteristics and modeling experience. The final values were selected based on a statistical evaluation of the performance measures. Twenty of the most influential parameters were selected for calibration. These parameters are related to surface runoff (CN2, CNCOEF, SURLAG, OV_N, and CH_N), evapotranspiration (ESCO), soil water (SOL_AWC and SOL_K), groundwater (GW_DELAY, GWQMN, ALPHA_BF, REVAPMN, and GW_REVAP), and reservoir operation (RES_ESA, RES_EVOL, RES_PSA, RES_PVOL, RES_VOL, RES_K, and EVRSV) processes." As shown below, adjusted parameter values and definitions were included in Table 1 of Chung et al (2017).

8. Results and discussion generally is redundant. This part need to be condensed. Some information can be incorporated in supplementary materials.

* Response: (Lines 460-563) Following the reviewer's suggestion, the manuscript has been revised overall and have we removed duplicate information as much as possible to condense 3. Results and discussion.

9. Page 10 line 237: "T-N was between 0.46 and" There should be a space between "0.46" and "and".

* Response: (Line 278) We added a space between "0.46" and "and".

10. Page 10 line 239: should there have a space before and after => ?

* Response: (Line 280) We added a space between before and after $\geq$.

11. Page 19 line 478: Please improve wording of the first sentence.

* Response: (Lines 523-525) We revised this as follows: "Figure 11 shows the poor watershed health of hydrology (Figure 11a), water quality (Figure 11b), and overlay results (Figure 11c) of a combination of both."
12. Conclusion did not interpolate researching findings well. The results showed the watershed health declined and targeted the vulnerable areas, but what is boarder impacts of these results? How will it be beneficial for watershed management? It would be more meaningful if authors can incorporate this information.

* Response: (Lines 601-606) We added new sentences about impacts of study results and beneficial effects for watershed management in Conclusion section as follows: "By listing all the information of the watershed health assessment, we can find vulnerable parts or healthy parts in the desired area and can provide basic data for action. The effectiveness of the watershed health that were evaluated in this study would be of reliable information because this approach entirely physically based. This approach can be utilized in a number of standard watersheds, local communities, and regions throughout the Han River basin and could be practically implemented in the watershed as a comprehensive watershed management plan by the government authorities or representative stakeholder."

13. What is limitation of this study, such as water quantity, quality data, or model input limitations? How to improve it in the further study? What kind of take-home messages you would like to delivery to readers?

* Response: (Lines 607-612) We added new sentences about limitation of water quantity, quality data, and model input in Conclusion section as follows: "Finally, the limitations of this study include the simulation of the water quantity and quality data for a possible long term changes in the watershed model. Although the prediction of long-term water quantity and quality data using the modeling is essential to assess water resource systems, the hydrologic and water quality conditions cannot be projected perfectly due to uncertainties in the models, climate data and other inputs required for the simulations. However, the results of this study are useful in terms of identifying potential watershed health issues regarding ongoing watershed change."

Please also note the supplement to this comment:

http://www.hydrol-earth-syst-sci-discuss.net/hess-2017-88/hess-2017-88-AC1-supplement.pdf

[Figure]

[Figure]

**Figure 1 Comparison of the observed and SWAT-simulated daily dam inflow during the
calibration (2005–2009) and validation (2010–2014) periods at (a) HSD, (b) SYD, (c) CJD, (d)
KCW, (e) YJW, (f) IPW, and (c) PDD.**

**Fig. 1.**

[Figure]

Figure 2 Comparison of the observed and SWAT-simulated daily sediment during the calibration (2005–2009) and validation (2010–2014) periods at (a) SG, (b) CSG, (c) JW, (d) KCW, (e) YJW, (f) IPW, and (c) PDD.

**Fig. 2.**

[Figure]

**Figure 3 Comparison of the observed and SWAT-simulated daily T-N during the calibration (2005–2009) and validation (2010–2014) periods at (a) SG, (b) CSG, (c) JW, (d) KCW, (e) YJW, (f) IPW, and (c) PDD.**

**Fig. 3.**

[Figure]

Figure 4 Comparison of the observed and SWAT-simulated daily T-P during the calibration
(2005–2009) and validation (2010–2014) periods at (a) SG, (b) CSG, (c) JW, (d) KCW, (e) YJW,
(f) IPW, and (c) PDD.

**Fig. 4.**

Table 1. Descriptions of calibrated parameters in Soil and Water Assessment Tool (SWAT) [32].

| Parameter | | Definition | Range | Adjusted Value (Average) | |
|---|---|---|---|---|---|
| | | | | Dams | Weirs |
| Surface runoff | CN2 | SCS curve number for moisture conditions | 35–98 | +12.5 | +7 |
| | CNCOEF | Plant ET curve number coefficient | 0.5–2 | 2 | 2 |
| | SURLAG | Surface runoff lag coefficient | 1–24 | 4 | 4 |
| | OV_N | Manning's "n" value for overland flow | 0.01–30 | 0.14 | 0.14 |
| | CH_N(1) | Manning's "n" value for tributary channels | 0.01–30 | 0.014 | 0.014 |
| Evapotranspiration | ESCO | Soil evaporation compensation coefficient | 0–1 | 0.9125 | 0.95 |
| Soil water | SOL_AWC | Available water capacity | 0–1 | 0.135 | 0.14 |
| | SOL_K | Saturated hydraulic conductivity (mm/hr) | 0–2000 | 25.8 | 25.8 |
| Ground water | GW_DELAY | Delay time for aquifer recharge (days) | 0–500 | 29 | 31 |
| | GWQMN | Threshold water level in a shallow aquifer for baseflow (mm) | 0–5000 | 1375 | 1000 |
| | ALPHA_BF | Baseflow recession constant | 0–1 | 0.725 | 0.048 |
| | REVAPMN | Threshold water level in a shallow aquifer for "revap" (mm) | 0–1000 | 750 | 750 |
| | GW_REVAP | Groundwater "revap" coefficient | 0.02–0.2 | 0.02 | 0.02 |
| Reservoir | RES_ESA | Reservoir surface area of the emergency spillway (km$^2$) | - | 48.25 | 4 |
| | RES_EVOL | Volume of water needed to fill the reservoir storage Volume of the emergency spillway (10$^6$ m$^3$) | - | 1495.25 | 13.667 |
| | RES_PSA | Reservoir surface area of the principal spillway (km$^2$) | - | 43 | 3 |
| | RES_PVOL | Reservoir storage volume of the principal spillway (10$^6$ m$^3$) | - | 1257.25 | 11.33 |
| | RES_VOL | Initial reservoir volume (10$^6$ m$^3$) | - | 674.75 | 9 |
| | RES_K | Hydraulic conductivity of the reservoir bottom (mm/hr) | 0–1 | 0.2 | 0.3 |
| | EVRSV | Lake evaporation coefficient | 0–1 | 0.525 | 0.6 |

**Fig. 5.**

---

## Author Comment (AC2) · 25 May 2017

MS No.: hess-2017-88

MS Type: Research article

Special Issue: Coupled terrestrial-aquatic approaches to watershed-scale water resource sustainability

Title: Title: Assessment of Integrated Watershed Health based on Natural Environment, Hydrology, Water Quality, and Aquatic Ecology

Journal: Hydrology and Earth System Sciences

Anonymous Referee #2

COMMENTS: This study assesses health condition of the Han River basin in South Korea based on monitoring data, water quantity and quality time series simulations of the SWAT model and an ensemble of indicators related to 6 components of the watershed landscape related to stream geomorphology, hydrology, water quality, aquatic habitat condition, and biological condition. The paper deals with an interesting topic which is watershed health condition. Indeed, there is a weak understanding of the complex processes and watershed components interactions that govern the healthy/unhealthy state of the watershed and such paper is needed to bridge the gap. This is a nice paper, well written and structured in a coherent way. But to my opinion, the approach needs to be improved by including an uncertainty assessment/analysis of the SWAT model. Authors used SWAT model simulations for water quality and quantity time series reconstruction which in-turn were used for indicators and sub-index development, as stated in the first specific object of the paper. Rely on model simulation for developing these indicators may add uncertainty in the indicators and sub-indexes. In addition, the definition of the reference condition here is crucial and used as a kind of "threshold" to discriminate between healthy and unhealthy watershed condition. This choice is based on SWAT simulation without any uncertainty analysis. I would prefer to see an acceptable range of reference condition based on model uncertainty analysis rather a single value of reference indicator.

1. lines 314-316: Authors mentioned that surface water and lateral groundwater flow interactions were of major importance for the water balance in the Han River basin. In particular, infiltration, return flow, groundwater recharge were important factors for the whole hydrological cycle. These results were based on SWAT simulations. Again, in absence of model uncertainty analysis the contribution of these components to the total water balance may vary or change depending on the parameter of the model. Therefore, I don't think that metrics developed based on the above results can be used

for establishing specific management objectives as stated by the authors in line 323.

* Response:

(Lines 247-259) We added a new paragraph in 2.5.2 Calibration and validation of the model section as follows: "In this study, uncertainty analysis was performed for the hydrology using daily dam inflow using the SUFI-2 method. This method was chosen because of their applicability to both simple and complex hydrological models. SUFI-2 is convenient and easy to implement and widely used in hydrology (e.g., Freer et al., 1996; Cameron et al., 2000; Blazkova et al., 2002). In SUFI-2, parameter uncertainty accounts for all sources of uncertainty, e.g., input uncertainty, conceptual model uncertainty, and parameter uncertainty (Gupta et al., 2005). The degree to which uncertainties are accounted for is quantified by a measure referred to as the P factor, which is the percentage of measured data bracketed by the 95% prediction uncertainty (95PPU). Another measure quantifying the strength of a calibration or uncertainty analysis is the R factor which is the average thickness of the 95PPU band divided by the standard deviation of the measured data. The excellence of calibration and prediction uncertainty is judged on the basis of the closeness of the P factor to 1 and the R factor to 0. For the uncertainty analysis, 20 parameters were selected by sensitivity analysis. In this study, three iterations were performed with 1,300 (100+200+1,000) model runs in each iteration. The coverage of measurements (P factor) and the average thickness (R factor) of the 95PPUs for model predictions were 0.79 and 0.32 for the dam inflow during calibration and validation periods." (Lines 281-284) We added NSE with inverse discharge (1/Q) in Table 2. We added new sentences: "Additionally, model calibration and validation included the NSE with inverse discharge (1/Q) for low flow. The average NSE with inverse discharge (1/Q) during the calibration (2005–2009) and validation (2010–2014) periods was 0.35 at HSD, 0.53 at SYD, 0.30 at CJD, 0.54 at KCW, 0.47 at YJW, 0.69 at IPW, and 0.58 at PDD."

(Lines 607-612) We added new sentences about limitation of water quantity, quality data, and model input in Conclusion section as follows: "Finally, the limitations of this

[Figure]

study include the simulation of the water quantity and quality data for a possible long term changes in the watershed model. Although the prediction of long-term water quantity and quality data using the modeling is essential to assess water resource systems, the hydrologic and water quality conditions cannot be projected perfectly due to uncertainties in the models, climate data and other inputs required for the simulations. However, the results of this study are useful in terms of identifying potential watershed health issues regarding ongoing watershed change."

We agree with your opinion. We know that the model is involved uncertainty, we tried to simulate spatial trends of water quantity and quality as successful as possible. The indicator score for the hydrology metric was re-scaled to normalize each sub-index score to a range from 0 to 1 using the percentile rank method. This index score shows the relative results for each standard watershed of the study area by calculating the various hydrologic components by the reference condition.

Please also note the supplement to this comment:
http://www.hydrol-earth-syst-sci-discuss.net/hess-2017-88/hess-2017-88-AC2-supplement.pdf

---

## Author Response (AR1)

**MS No.:** hess-2017-88
**MS Type:** Research article
**Special Issue:** Coupled terrestrial-aquatic approaches to watershed-scale water resource sustainability
**Title:** Assessment of Integrated Watershed Health based on the Natural Environment, Hydrology, Water Quality, and Aquatic Ecology
**Journal:** Hydrology and Earth System Sciences

**Anonymous Referee #1**

**COMMENTS:** This study evaluated health condition of a watershed of the Han River basin (34,148 km2) in South Korea was performed using monitoring data and SWAT modeling results. Six essential indicators of healthy watersheds were used in the assessment: landscape condition, geomorphology, hydrology, water quality, habitat, and biological condition. The research findings from this study provided guidance for watershed management at the watershed scale based on specific management objectives and can combined with any of the other sub-indices in the Han River basin for use in determining priority conservation areas. This paper is well organized and well written generally. Detailed method description was incorporated. The scientific results and conclusions were presented in a clear, concise, and well-structured way. The number and quality of references is appropriate. But method and results should be reduced. The importance of six essential indicators of healthy watersheds was not well described. More in-depth discussion should be included to support the interpretations and conclusions. This manuscript can be reviewed after major revisions. What is the novel idea this manuscript provided to scientific knowledge? Please describe it and use your results and discussion to support it.

**General**

1. The last sentence of the abstract "The results suggest that approaches aimed at simultaneously improving the water quality, hydrology, and aquatic ecology conditions may be necessary to improve integrated watershed health." Is this the scientific questions being answered in this manuscript? Please provide specific discussion of results and summarize them in conclusion to support this point. Otherwise, I do not think this sentence should be here.

   ● Response:
   (Lines 27-32) We removed the last sentence of the abstract and revised this section as follows: "The results indicate that the watershed's health declined during the most recent ten-year period of 2005–2014, as indicated by the worse results for the surface process metric and soil water dynamics compared to those of the 1995–2004 period. The integrated watershed health tended to decrease farther downstream within the watershed."

2. 2.4 Hydrology and water quality simulations using the SWAT model: the session is mainly focus on basic information about SWAT. If it is not specific for your project, it is better to put information in the Introduction rather than in Methods. And authors already described data collection related to SWAT model setup and SWAT outputs in 2.3, thus it is better to introduce SWAT model before discussing data related to it.

   ● Response:
   (Lines 153-162, and 199-206) Section 2.4, "Hydrology and water quality simulations using the SWAT model", mainly focused on both basic information regarding the SWAT and the model calibration and validation for the hydrology and water-quality simulation data. The information in this section is very important for the watershed-health assessment. We added a new section 2.3, "SWAT model description", before section 2.4, "Data collection", and removed the basic information regarding the SWAT in section 2.5, "Hydrology and water quality simulations with the SWAT model".

3.   Is 90 m grid size DEM data sufficient to accurately simulate hydrology and water quality at such a large area? Is there any higher resolution elevation data can be used?

● Response:

(Lines 212-213) Our study area included portions of North Korea. We had a 30-m DEM that covered South

Korea, but we did not have data in North Korea. Therefore, we used a 90-m global DEM from the Shuttle

Radar Topography Mission (SRTM) of the International Centre for Tropical Agriculture (CIAT). As shown

Figures 1, 2, 3, and 4 below, the results for the hydrology and water quality were reasonable. I believe that precipitation had an even greater effect on the hydrologic simulations than the DEM resolution did. In addition, the resolution of the 90-m DEM was appropriate to simulate the watershed's hydrology for the 237

sub-watersheds (average area of 144 km²) by using the SWAT model.

[Figure]

**Figure 1 Comparison of the observed and SWAT-simulated daily dam inflow during the calibration (2005–2009) and validation (2010–2014) periods at (a) HSD, (b) SYD, (c) CJD, (d) KCW, (e) YJW, (f) IPW, and (g) PDD.**

[Figure]

**Figure 2 Comparison of the observed and SWAT-simulated daily sediment during the calibration (2005–2009) and validation (2010–2014) periods at (a) SG, (b) CSG, (c) JW, (d) KCW, (e) YJW, (f) IPW, and (g) PDD.**

[Figure]

**Figure 3 Comparison of the observed and SWAT-simulated daily T-N during the calibration (2005–2009) and validation (2010–2014) periods at (a) SG, (b) CSG, (c) JW, (d) KCW, (e) YJW, (f) IPW, and (g) PDD.**

[Figure]

**Figure 4 Comparison of the observed and SWAT-simulated daily T-P during the calibration (2005–2009) and validation (2010–2014) periods at (a) SG, (b) CSG, (c) JW, (d) KCW, (e) YJW, (f) IPW, and (g) PDD.**

4.   Is calibration period (2005–2009) and validation period (2010–2014) both incorporate wet and dry years?

● Response:
(Lines 249-254) We incorporated both wet and dry years in the calibration (2005–2009) and validation
periods (2010–2014). The average annual precipitation of the Han River basin is 1,300 mm. For the
calibration period (2005–2009), the wet and dry years were 2006 (1,625 mm) and 2008 (1,160 mm). For the
validation period (2010–2014), the wet and dry years were 2011 (1,640 mm) and 2014 (734 mm).

5.   Statistical evaluation criteria R2, NSE and PBIAS are all sensitive to high values. Criteria less sensitive to
high values, such as Modified NSE and KGE, may could be incorporated.

● Response:
(Lines 291-293) We added the NSE with inverse discharge (1/Q) in Table 2. We also added the following
sentences: "Additionally, the model calibration and validation included the NSE with inverse discharge (1/Q)
for low flow. The average NSE with inverse discharge (1/Q) during the calibration (2005–2009) and
validation (2010–2014) periods was 0.35 at HSD, 0.53 at SYD, 0.30 at CJD, 0.54 at KCW, 0.47 at YJW,
0.69 at IPW, and 0.58 at PDD."

6.   Page 8 line 197: this paragraph described a lot of detailed information about dams. It is better to condense
it and save more space for in-depth discussions. How was dam information being set in SWAT model?

● Response:
(Lines 218-226 and 230-243) We removed the paragraph that described the dam informations. We added the
following sentences regarding the dam information that was set in the SWAT model: "The flow and water
quality of the Han River are affected by the discharge operations of these large dams and weirs; therefore,
dam and weir operations must be incorporated into the modeling framework to enable successful modeling.
In the SWAT model, dam operations are modeled based on measured daily discharges, measured monthly
discharges, average annual discharges, or target storage volumes. In this study, the measured daily
discharges from the four dams and three weirs were directly imported into the SWAT model."

7.   Page 9 line 226: "The calibrated parameters and hydrograph of the calibration results in the Han River basin
were described by 227 Chung et al (2017)." Parameter definition, physical meaning, range used for
calibration and calibrated values are very important information. Please describe this information in
supplementary materials to prove that your calibration and validation is reliable.

● Response:
(Lines 269-276) We added the following sentences regarding information for the parameter definitions and
physical meanings: "In this study, both calibration and validation were manually performed by using a trial-
and-error approach within recommended ranges to maximize the expert knowledge of watershed
characteristics and modeling experience. The final values were selected based on a statistical evaluation of
the performance measures. Twenty of the most influential parameters were selected for calibration. These
parameters are related to surface runoff (CN2, CNCOEF, SURLAG, OV_N, and CH_N), evapotranspiration
(ESCO), soil water (SOL_AWC and SOL_K), groundwater (GW_DELAY, GWQMN, ALPHA_BF,
REVAPMN, and GW_REVAP), and reservoir operation (RES_ESA, RES_EVOL, RES_PSA, RES_PVOL,
RES_VOL, RES_K, and EVRSV) processes."
As shown below, the adjusted parameter values and definitions were included in Table 1 from Chung et al.
(2017).

Table 1. Descriptions of calibrated parameters in Soil and Water Assessment Tool (SWAT) [32].

| Parameter | | Definition | Range | Adjusted Value (Average) | |
|---|---|---|---|---|---|
| | | | | Dams | Weirs |
| Surface runoff | CN2 | SCS curve number for moisture conditions | 35–98 | +12.5 | +7 |
| | CNCOEF | Plant ET curve number coefficient | 0.5–2 | 2 | 2 |
| | SURLAG | Surface runoff lag coefficient | 1–24 | 4 | 4 |
| | OV_N | Manning's "n" value for overland flow | 0.01–30 | 0.14 | 0.14 |
| | CH_N(1) | Manning's "n" value for tributary channels | 0.01–30 | 0.014 | 0.014 |
| Evapotranspiration | ESCO | Soil evaporation compensation coefficient | 0–1 | 0.9125 | 0.95 |
| Soil water | SOL_AWC | Available water capacity | 0–1 | 0.135 | 0.14 |
| | SOL_K | Saturated hydraulic conductivity (mm/hr) | 0–2000 | 25.8 | 25.8 |
| Ground water | GW_DELAY | Delay time for aquifer recharge (days) | 0–500 | 29 | 31 |
| | GWQMN | Threshold water level in a shallow aquifer for baseflow (mm) | 0–5000 | 1375 | 1000 |
| | ALPHA_BF | Baseflow recession constant | 0–1 | 0.725 | 0.048 |
| | REVAPMN | Threshold water level in a shallow aquifer for "revap" (mm) | 0–1000 | 750 | 750 |
| | GW_REVAP | Groundwater "revap" coefficient | 0.02–0.2 | 0.02 | 0.02 |
| Reservoir | RES_ESA | Reservoir surface area of the emergency spillway ($km^2$) | - | 48.25 | 4 |
| | RES_EVOL | Volume of water needed to fill the reservoir storage Volume of the emergency spillway ($10^6$ $m^3$) | - | 1495.25 | 13.667 |
| | RES_PSA | Reservoir surface area of the principal spillway ($km^2$) | - | 43 | 3 |
| | RES_PVOL | Reservoir storage volume of the principal spillway ($10^6$ $m^3$) | - | 1257.25 | 11.33 |
| | RES_VOL | Initial reservoir volume ($10^6$ $m^3$) | - | 674.75 | 9 |
| | RES_K | Hydraulic conductivity of the reservoir bottom (mm/hr) | 0–1 | 0.2 | 0.3 |
| | EVRSV | Lake evaporation coefficient | 0–1 | 0.525 | 0.6 |

8. Results and discussion generally is redundant. This part need to be condensed. Some information can be incorporated in supplementary materials.

● Response:
(Lines 483-614) Following the reviewer's suggestion, the manuscript has been generally revised, and we removed duplicate information as much as possible to condense section 3, "Results and discussion".

9. Page 10 line 237: "T-N was between 0.46 and" There should be a space between "0.46" and "and".

● Response:
(Line 288) We added a space between "0.46" and "and".

10. Page 10 line 239: should there have a space before and after => ?

● Response:
(Line 290) We added a space before and after ≥.

11. Page 19 line 478: Please improve wording of the first sentence.

● Response:

(Lines 552-554) We revised this sentence as follows: "Figure 11 shows the poor watershed health in terms
of the hydrology (Figure 11a), water--quality (Figure 11b), and overlay results (Figure 11c)."

12. Conclusion did not interpolate researching findings well. The results showed the watershed health declined
and targeted the vulnerable areas, but what is boarder impacts of these results? How will it be beneficial for
watershed management? It would be more meaningful if authors can incorporate this information.

● Response:
(Lines 633-638) We added the following sentences regarding the effects of the study results and beneficial
effects for watershed management in the Conclusions section: "Listing all the information of the watershed-
health assessment can indicate vulnerable or healthy regions in the desired area and can provide basic data
for action. The effectiveness of the watershed--health evaluation in this study can produce reliable
information because this approach is entirely physically based. This approach can be utilized in a number
of standard watersheds, local communities, and regions throughout the Han River basin and can be
practically implemented in the watershed as a comprehensive watershed management plan by government
authorities or representative stakeholders."

13. What is limitation of this study, such as water quantity, quality data, or model input limitations? How to
improve it in the further study? What kind of take-home messages you would like to delivery to readers?

● Response:
(Lines 639-644) We added the following sentences regarding the limitations of this study in terms of the
water quantity, quality data, and model input in the Conclusions section: "Finally, the limitations of this
study include the simulation of water quantity and quality data for possible long-term changes in the
watershed model. Although the prediction of long-term water quantity and quality data with this modeling
is essential to assess water-resource systems, the hydrologic and water quality conditions cannot be perfectly
projected because of uncertainties in the models, climate data and other inputs that are required for the
simulations. However, the results of this study are useful in terms of identifying potential watershed--health
issues regarding ongoing watershed changes."

**MS No.:** hess-2017-88
**MS Type:** Research article
**Special Issue:** Coupled terrestrial-aquatic approaches to watershed-scale water resource
sustainability
**Title:** Assessment of Integrated Watershed Health based on the Natural Environment, Hydrology,
Water Quality, and Aquatic Ecology
**Journal:** Hydrology and Earth System Sciences
**Anonymous Referee #2**
**COMMENTS:** This study assesses health condition of the Han River basin in South Korea based on monitoring
data, water quantity and quality time series simulations of the SWAT model and an ensemble of indicators related to
6 components of the watershed landscape related to stream geomorphology, hydrology, water quality, aquatic habitat
condition, and biological condition. The paper deals with an interesting topic which is watershed health condition.
Indeed, there is a weak understanding of the complex processes and watershed components interactions that govern
the healthy/unhealthy state of the watershed and such paper is needed to bridge the gap. This is a nice paper, well
written and structured in a coherent way. But to my opinion, the approach needs to be improved by including an
uncertainty assessment/analysis of the SWAT model.
Authors used SWAT model simulations for water quality and quantity time series reconstruction which in-turn were
used for indicators and sub-index development, as stated in the first specific object of the paper. Rely on model
simulation for developing these indicators may add uncertainty in the indicators and sub-indexes. In addition, the
definition of the reference condition here is crucial and used as a kind of "threshold" to discriminate between healthy
and unhealthy watershed condition. This choice is based on SWAT simulation without any uncertainty analysis. I
would prefer to see an acceptable range of reference condition based on model uncertainty analysis rather a single
value of reference indicator.

**General**

1. lines 314-316: Authors mentioned that surface water and lateral groundwater flow interactions were of major
importance for the water balance in the Han River basin. In particular, infiltration, return flow, groundwater
recharge were important factors for the whole hydrological cycle. These results were based on SWAT
simulations. Again, in absence of model uncertainty analysis the contribution of these components to the
total water balance may vary or change depending on the parameter of the model. Therefore, I don't think
that metrics developed based on the above results can be used for establishing specific management
objectives as stated by the authors in line 323.

● Response:
(Lines 255-268) We added a new paragraph in section 2.5.2, "Calibration and validation of the model": "In
this study, uncertainty analysis was performed for the hydrology by using the daily dam inflow with the
SUFI-2 method. This method was chosen because of its applicability to both simple and complex
hydrological models. SUFI-2 is convenient and easy to implement and widely used in hydrology (e.g., Freer
et al., 1996; Cameron et al., 2000; Blazkova et al., 2002). In SUFI-2, parameter uncertainty considers all
sources of uncertainty, e.g., input uncertainty, conceptual model uncertainty, and parameter uncertainty
(Gupta et al., 2005). The degree to which uncertainties are considered is quantified by a measure called the
P factor, which is the percentage of the measured data that are bracketed by the 95% prediction uncertainty
(95PPU). Another measure that quantifies the strength of a calibration or uncertainty analysis is the R factor,
which is the average thickness of the 95PPU band divided by the standard deviation of the measured data.
The excellence of the calibration and prediction uncertainty is judged based on the closeness of the P factor
to 1 and the closeness of the R factor to 0. Twenty parameters were selected by sensitivity analysis for the
uncertainty analysis. In this study, three iterations were performed with 1,300 (100+200+1,000) model runs
in each iteration. The coverages of the measurements (P factor) and the average thickness (R factor) of the
95PPUs for the model predictions were 0.79 and 0.32, respectively, for the dam inflow during the calibration
and validation periods."

(Lines 291-293) We added the NSE with inverse discharge (1/Q) in Table 2. We also added the following sentences: "Additionally, the model calibration and validation included the NSE with inverse discharge (1/Q) for low flow. The average NSE with inverse discharge (1/Q) during the calibration (2005–2009) and validation (2010–2014) periods was 0.35 at HSD, 0.53 at SYD, 0.30 at CJD, 0.54 at KCW, 0.47 at YJW, 0.69 at IPW, and 0.58 at PDD."

(Lines 639-644) We added new sentences about limitation of water quantity, quality data, and model input in Conclusion section as follows: "Finally, the limitations of this study include the simulation of water quantity and quality data for possible long-term changes in the watershed model. Although the prediction of long-term water quantity and quality data with this modeling is essential to assess water-resource systems, the hydrologic and water quality conditions cannot be perfectly projected because of uncertainties in the models, climate data and other inputs that are required for the simulations. However, the results of this study are useful in terms of identifying potential watershed-health issues regarding ongoing watershed changes." We agree with your opinion. We recognized that the model involves uncertainty, so we attempted to simulate the spatial trends of the water quantity and quality. The indicator score for the hydrology metric was re-scaled to normalize each sub-index score to a range from 0 to 1 by using the percentile rank method. This index score shows the relative results for each standard watershed of the study area by calculating the various hydrologic components according to the reference condition.

**Assessment of Integrated Watershed Health based on the Natural Environment, Hydrology, Water Quality, and Aquatic Ecology**

So Ra Ahn[a] and Seong Joon Kim[b]

[a]Assistant Research Scientist (Ahn), Texas A&M AgriLife Research Center at El Paso, Texas 79927, USA; and [b]Professor (Kim), Department of Civil and Environmental System Engineering, Konkuk University, Seoul 05029, South Korea, Email: kimsj@konkuk.ac.kr

**Abstract**

The watershed health, including the natural environment, hydrology, water quality, and aquatic ecology, is assessed for the Han River basin (34,148 km²) in South Korea by using the Soil and Water Assessment Tool (SWAT). The evaluation procedures follow those of the Healthy Watersheds Assessment by the U.S. Environmental Protection Agency (EPA). Six components of the watershed landscape are examined to evaluate the watershed health (basin natural capacity): stream geomorphology, hydrology, water quality, aquatic habitat condition, and biological condition. In particular,  the SWAT is applied to the study basin for the hydrology and water quality components, including  237 sub-watersheds (within a standard watershed on the Korea Hydrologic Unit Map) along with three multipurpose dams, one hydroelectric dam, and three multifunction weirs. The SWAT is calibrated (2005–2009) and validated (2010–2014) by using each dam and weir operation, the flux-tower evapotranspiration, the time-domain reflectometry (TDR soil moisture, and groundwater level data for the hydrology assessment and by using sediment, total phosphorus, and total nitrogen data for the water quality assessment. The water balance, which consider the surface–groundwater interactions and  variations in the stream-water quality, are  quantified according to the sub-watershed-scale relationship between the watershed hydrologic cycle and stream-water quality. We assess the integrated watershed health according to the U.S. EPA evaluation process based on the vulnerability levels of the natural environment, water resources, water quality, and ecosystem components.

The results indicate that the watershed's health declined during the most recent ten-year period of 2005–2014, as indicated by the worse results for the surface process metric and soil water dynamics compared to those of the 1995–2004 period. The integrated watershed health tended to decrease farther downstream within the watershed.

Keywords: Watershed health assessment; SWAT; Watershed hydrology; Water quality; Aquatic ecology

**1. Introduction**

Watershed management can be defined as the integrated and iterative decision process  applied to maintain the sustainability of resources through the balanced use and conservation of water quantity, land, vegetation, and other natural resources within the watershed. Rivers   a constituent element of  watershed ecosystem that  of primary concern for watershed management;  river discharge and water quality are key components of  watershed ecosystem, and their interactions can be affected by land use and vegetation cover. The Han River basin in South Korea, with its large-scale water supply dams and weirs, is a rare case . Twenty-six years ago, the government initiated programs  to restore the environmental and human health-related quality of the Han River basin. However, an integrated approach that considers the water supply, water-quality improvement, and natural-ecosystem maintenance and their interactions within the watershed has been lacking. A broader view of watershed ecosystems is essential  to truly protect the chemical, physical, and biological integrity of our watersheds (U.S. EPA, 2012).

One of the key components of watershed-management strategies is to increase the protection of healthy waters, including healthy watersheds. A key component of watershed health is its ability to withstand, recover from, or adapt to disturbances, such as floods and droughts. A more complete understanding of the watershed-ecosystem components that affect watershed health is important to identify management actions to protect healthy watersheds. Without an integrated watershed-health-assessment system, any successes in restoring impaired waters will be limited and the many socioeconomic benefits of healthy watershed systems will be lost.

Generally, the assessment of the major components of watershed health must incorporate evaluations of the natural environment, hydrology, water quality and aquatic ecology. A number of studies have recently assessed the potential for effective watershed management through an analysis of a variety of health indicators. Sanchez et al. (2015) characterized the relationships among in-stream health indicators (flow, sediment, and nutrient loads) by using the Soil and Water Assessment Tool (SWAT)  and the socioeconomic measures of communities by using spatial-clustering techniques and confirmatory-factor analysis in the Saginaw River watershed in Michigan. Cook et al. (2015) examined these relationships in five watersheds along the Virginia–Kentucky border and explored the effects of both the water quality and habitat on benthic macroinvertebrates by using the data from a three-year field study and Virginia Stream Condition Index (VSCI) scores to evaluate site-specific environmental variables (land use, habitat metrics, and water-quality parameters),

. Tango and Batiuk (2016) analyzed the interactions that affect the watershed and bay water-quality recovery 
[revised manuscript text omitted]
. The flow and water quality of the Han River are affected by the discharge operations of these large dams and weirs; therefore, dam and weir operations must be incorporated into the modeling framework to enable successful modeling. In the SWAT model, dam operations are modeled based on measured daily discharges, measured monthly discharges, average annual discharges, or target storage volumes. In this study, the measured daily discharges from the four dams and three weirs were directly imported into the SWAT model.

For the calibration and validation of the stream water quality, ten years (2005–2014) of eight-day intervals for sediments, T-N, and T-P data were obtained from seven hydrology stations (SG, CSG, JW, KCW, YJW, IPW, and PDD) that arefor the hydrology monitored by the KME. Figure 2a shows the gauging stations for the SWAT modeling.

2.45.2 Calibration and validation of the model

The SWAT model was calibrated at seven locations in the main river reaches by using five years (2005–2009) of daily inflow, storage volume data for the dams and weirs, sediments, T-N, and T-P data and was subsequently validated by using another five years (2010–2014) of data withusing the average calibrated parameters. In addition, the model was spatially calibrated and validated by using evapotranspiration and soil moisture data that were measured at two locations (SM and CM) and groundwater level data that were measured at five locations (GPGP, YPGG, YPYD, YIMP, and HCGD) over five years (2009–2013).

In this study, uncertainty analysis was performed for the hydrology by using the daily dam inflow using the SUFI-2 method. This method was chosen because of its applicability to both simple and complex hydrological models. SUFI-2 is convenient and easy to implement and widely used in hydrology (e.g., Freer et al., 1996; Cameron et al., 2000; Blazkova et al., 2002). In SUFI-2, parameter uncertainty considers all sources of uncertainty, e.g., input uncertainty, conceptual model uncertainty, and parameter uncertainty (Gupta et al., 2005). The degree to which uncertainties are considered is quantified by a measure called the P factor, which is the percentage of the measured data that are bracketed by the 95% prediction uncertainty (95PPU). Another measure that quantifies the strength of a calibration or uncertainty analysis is the R factor, which is the average thickness of the 95PPU band divided by the standard deviation of the measured data. The excellence of calibration and prediction uncertainty is judged based on the closeness of the P factor to 1 and the closeness of the R factor to 0. Twenty parameters were selected by sensitivity analysis for the uncertainty analysis. In this study, three iterations were performed with 1,300 (100+200+1,000) model runs in each iteration. The coverages of the measurements (P factor) and the average thickness (R factor) of the 95PPUs for the model predictions were 0.79 and 0.32, respectively, for the dam inflow during the calibration and validation periods.

In this study, both calibration and validation were manually performed by using a trial-and-error approach within recommended ranges to maximize the expert knowledge of watershed characteristics and modeling experience. The final values were selected based on a statistical evaluation of the performance measures. Twenty of the most influential parameters were selected for calibration. These parameters are related to surface-runoff (CN2,

CNCOEF, SURLAG, OV_N, and CH_N), evapotranspiration (ESCO), soil-water (SOL_AWC and SOL_K), groundwater (GW_DELAY, GWQMN, ALPHA_BF, REVAPMN, and GW_REVAP), and reservoir-operation (RES_ESA, RES_EVOL, RES_PSA, RES_PVOL, RES_VOL, RES_K, and EVRSV) processes. The calibrated parameters and hydrograph of the calibration results in the Han River basin were described by Chung et al. (2017).

The statistical results for the hydrology and water quality for the model calibration and validation are summarized in Table 2. The coefficient of determination ($R^2$), the Nash and Sutcliffe model efficiency (NSE), the root-mean-square error (RMSE), and the percent bias (PBIAS) were used to evaluate the ability of the SWAT model to replicate temporal trends in the observed hydrological and water quality data. The $R^2$ value for the dam inflow was greater than 0.59. The average NSE was 0.59 at HSD, 0.78 at SYD, 0.61 at CJD, 0.79 at KCW, 0.77 at YJW, 0.88

at IPW, and 0.87 at PDD. The PBIAS values of HSD, CJD, SYD, KCW, YJW, IPW and PDD were 13.5%, 12.2%,

9.4%, 11.5%, 19.8%, 21.4%, and 4.5%, respectively. The average $R^2$ for the dam-storage volume was between 0.40 and 0.96 and the PBIAS was between 0.9% and 18.9% for each calibration point. The average $R^2$ for evapotranspiration was between 0.70 and 0.81, that for the soil moisture was between 0.75

and 0.85, and that for the groundwater level was between 0.40 and 0.70 for each calibration point. The average $R^2$ for the sediment was between 0.54 and 0.90, that for the T-N was between 0.46 and 0.82, and that for the T-P was between

0.47 and 0.80 for each calibration point. The calibration results were consistent with the SWAT calibration guidelines (NSE ≥ 0.5, PBIAS ≤ 28%, and $R^2$ ≥ 0.6; Moriasi et al., 2007; Santhi et al., 2001) and were found to be satisfactory.

Additionally, the model calibration and validation included the NSE with inverse discharge (1/Q) for low flow. The average NSE with inverse discharge (1/Q) during the calibration (2005–2009) and validation (2010–2014) periods was 0.35 at HSD, 0.53 at SYD, 0.30 at CJD, 0.54 at KCW, 0.47 at YJW, 0.69 at IPW, and 0.58 at PDD.

<Table 2>

2.6 Data reconstruction for the watershed health assessment

2.6.1 Landscape condition

The area of natural land cover (forest, wetland, river, and natural grassland) within a watershed can be an important indicator of watershed health. Impervious land cover that is associated with roads and residential and urban areas can increase watershed runoff, leading to instream flow alteration, geomorphic instability, and increased pollutant loading.

According to previous studies, a smaller area of impervious land cover may significantly affect aquatic ecosystem health (e.g., King et al., 2011; Wang and Yin, 1997).

The extent and connectivity of the natural land cover within a watershed are very important for ecological integrity. Natural land cover within the watershed, and especially within headwater areas and riparian corridors, maintains the hydrologic regime, regulates the inputs of nutrients and organic matter, and provides habitats for fish and wildlife (U.S. EPA, 2012). In this study, assessing the connectivity of the natural land cover (forest, wetland, river, and natural grassland) of watersheds involved a green-area assessment; green areas comprise areas of unfragmented natural land cover and corridors of sufficient width to allow the migration of wildlife between the watersheds (Figure 3a). For the 237 sub-watersheds of the Han River basin, the percentage of each watershed area that was occupied by natural land cover (habitat blocks) was calculated by using GIS techniques. The green area metric was calculated as follows:

$Green\ area\ metric = \dfrac{Area\ (km^2)\ of\ natural\ land\ cover\ in\ watershed}{Total\ area\ (km^2)\ in\ watershed}$

(1)

The amount of natural land cover within the active river area is another important indicator of the landscape condition. The natural land cover within the active river area, including the river channel, lakes and ponds, and the riparian lands, is necessary for the physical and ecological functioning of the aquatic ecosystem (U.S. EPA, 2012).

Active river areas, in their natural state, maintain the ecological integrity of rivers, streams, and riparian areas and the connection of these areas to the local ground-water system (IPCC, 2007). The methods that are used to delineate the active river area involve GIS techniques and analyses of elevation, land-_-cover, and wetland data. For  streamside areas for which criteria have not yet been decided , an area with a width of 30–50 m can be used as a cutoff to identify streamside material contribution areas (U..S. EPA, 2012). In this study,

 the percentage of natural land cover within the riparian area within 50

m of the stream was calculated for the 237 sub-watersheds in the Han River basin by using GIS techniques (Figure 3b). The active river area metric was calculated as follows:

$$Active\ river\ area\ metric\ = \frac{Area(km^2)\text{-}\ of\ natural\ land\ cover\ in\ active\ river\ area}{Total\ area\ (km^2)\ in\ active\ river\ area}$$

(2)

<Figure 3>

2.6.2 Stream geomorphic condition

The natural stream geomorphology can be an important indicator of watershed health because it can fragment both the terrestrial and aquatic habitats throughout a watershed. Kline et al. (2009) performed detailed assessments of stream geomorphic conditions by using the Vermont Stream Geomorphic Assessment Protocols for  streams in

Vermont, USA. These assessment protocols are GIS-based analyses that use elevation, land cover, and stream network data layers to classify stream types and evaluate the conditions of individual reaches based on a comparison to reference conditions for that stream type.

Table 3 provides descriptions of the stream geomorphic condition  that are determined through the stream-_-impact rating and the stream order for the watershed-_-health assessment of the geomorphic condition in the

Han River basin. In this study, the  geomorphic condition was assessed in a  similar manner to wthat was used for the stream-condition categories of the Vermont Stream Geomorphic Assessment

Protocols. The stream order was calculated for nine levels (Figure 4a) by using a DEM and stream map, and four river classifications were created through follow-up analyses with detailed land-cover assessments (Figure 4b). FThere are four river classifications were used: for reference (mountainous river, stream order 1), good (small river, stream orders

2–3), fair (local river, stream orders 4–5), and poor (urban and national river, stream orders 6–9). The percentage of the assessed stream length in the reference condition was calculated for each watershed. The stream geomorphology metric was calculated as follows:

$$Stream\ geomorphology\ metric = \frac{Stream\ length\ (km)\ of\ reference\ condition\ in\ watershed}{Total\ stream\ length\ (km)\ in\ watershed}$$

(3)

<Figure 4>

<Table 3>

2.56.3 Hydrologic condition

The assessment of the hydrologic condition of a watershed requires long-term streamflow observation data for the 237

sub-watersheds of Han River basin. However, insufficientthere were not enough gauging stations were available to fully assess the entire watershed over the entirefull thirty-year period. NoThere were no data were available for the water-balance components that were associated with the surface–groundwater interactions, except for the streamflow.

Where unavailable, these long-term flow data are couldnot available, they can be estimated by using hydrologic modeling techniques. ThusTo 
[revised manuscript text omitted]
. The sampling areas that were used to explain the differences in the watershed- health results for each component were the standard watersheds 101206 (urban 1.4% and forest 88.1%), 100201 (urban 0.8% and forest 88.2%) and 101801 (urban 9.8% and forest 5%) (Figure 2a). The 101206, 100201, and 101801 standard watersheds are located in the upstream region of the Soyang Dam (SYD), in the upstream region of the Chungju Dam (CJD), and in the downstream region of the Paldang Dam (PDD), respectively.

  Figure 12a shows the sub-index scores for the watershed- health assessment  according to  two assessment indicators (Figure 3). The spatial patterns of the watershed health for green areas were healthier in upstream watersheds because the natural land cover was greater the farther the watersheds were from  urban area. The spatial patterns of the watershed health for the active river area within 50 m of a stream were healthier for the upstream watersheds for the same reason. For the 101206 standard watershed, the normalized values of the green area and the active river area were 0.93 and 0.82, respectively, and the sub-index score of 0.89, which integrated the two normalized values, indicated a very healthy watershed. For the 100201 standard watershed, the normalized values of the green area and the active river area were 0.78 and 0.57, respectively, and the sub-index score of .0.66, which integrated the two normalized values, indicates a less healthy watershed. In contrast, the 101801 standard watershed was revealed to be in very poor health, with a score of 0.17 for the sub-index, while the normalized values of the green area and active river area were 0.25 and 0.09, respectively. Hence, the study found that the downstream reaches of the Han River basin are in greater need of green areas and active river areas compared to the upstream reaches.

 Figure 12b shows the sub-index scores for the watershed- health assessment when using stream geomorphology indicators (Figure 4). The percentage of the length of the assessed stream channel in the reference condition was greater for the upstream watershed than for the downstream watershed. The high-gradient mountainous streams in the upstream watershed are characterized by relatively clean streams that have not been subject to land-cover modifications or river- improvement work.

The sub-index results of the hydrologic (Figure 5) and water-quality (Figure 6) conditions are shown in Figures 12c and d, respectively. The precipitation in the watershed directly affects the surface runoff and sediment transport and is the most important factor that affects the maintenance of the water quantity and can thus be used to identify critical areas for maintaining watershed health. Nutrient (T-N and T-P)

loads are often correlated with surface runoff and sediment transport rates (USDA-SCS, 1972). The fugitive sediment from the landscape is carried by overland flow (surface runoff), and the dominant pathway for nitrate loss is through leaching into groundwater and then via base flow (Randall and Mulla, 2001).

The sub-indices of the hydrologic condition that were calculated by the four hydrologic classifications, such as the total metric , surface process metric , soil water dynamics metric

, and groundwater dynamics metric , and the water quality condition that was calculated by the sediment, T-N, and T-P were split into three periods of ten years—1985–1994, 1995–2004, and

2005–2014 for the assess changes over time (Figure 9). The test areas that were used to explain the differences in the  watershed-health results for the hydrologic and water quality components were the

SYD  and CJD watershed  in the upstream region and the PDD  and lower watershed in the downstream region (Figure 2c). For the SYD watershed (Figure 9a), the watershed health scores of the surface water, soil water, and groundwater hydrology increased in the recent past compared to the period 1985–1994

because of the slight increases in PREC and TQ; thus, the watershed water quality decreased.

The health of the hydrology in the CJD watershed showed a decreasing tendency in contrast to the SYD watershed because of the decrease in PREC and TQ (Figure 9b). T

he groundwater of the PDD watershed was not sufficient, but the overall watershed-health scores for the

PDD and lower watersheds remained within their reference levels (approximately 0.5) (Figure 9c and d). This water- quantity stress (large volume of water in the stream) may have negatively aeffected thes on water quality, with a decreased watershed-health score for the sediment, T-N, and T-P. In particular, the SYD watershed was rich in soil water and the CJD watershed was rich in surface and groundwater.

Figure 10 shows the changes in the watershed-health index score changes for the hydrologic and water quality conditions during 1995–2004 and the most recent ten years (2005–2014) based on the reference period (1985–

1994). "Improved health", "deteriorating health", and "no change" area in the Han River basin are illustrated with green, red, and white, respectively. TOn the whole, the watershed's hydrologic condition was better in the North Han

River basin compared to the South Han River basin. In particular, during the last ten years (Figure 10b), the watershed's health was poorer because ofdue to worse results for the surface processes metric and soil water dynamics compared to those of the 1995–2004 period (Figure 10a). However, in the case of water quality, during the last ten years (Figure 10d), the watershed's health increasingly improved in portionsparts of the Han River basin compared to

1995–2004 (Figure 10c), while the water quality of the Chungju dam (CJD) watershed deterioratedwas growing worse.

The water-quality policy of South Korea, which was developed after years of hard work and high costs, thus resulted in some improvements.

Figure 11 shows the overlay results (Figure 11c) showing the poor watershed health of both hydrology (Figure 11a) and water quality (Figure 11b). Figure 11 shows the poor watershed health in terms of the hydrology (Figure 11a), water-quality (Figure 11b), and overlay results (Figure 11c) of a combination of both. The five poor levels for theof hydrology and water quality were calculated as the difference between (b) and (a) inof Figure 10 and between (d) and (c) inof Figure 10, respectively. The spatial distributions of the poor watershed-health levels enableallow us to understand the vulnerable areas in parts of the CJD watershed, the upstream SYD watershed, and the downstream PDD watershed with respect to the hydrology and water quality.

<Figure 9>

<Figure 10>

<Figure 11>

Figure 7 shows the aquatic habitat condition for the aquatic habitat connectivity (Figure 7a) and wetland (Figure 7b) indicators in the Han River basin. Figure 12e shows the sub-index scores for the watershed-health assessment  according to  two assessment indicators (Figure 7). The spatial-distribution patterns of the reservoirs for aquatic-habitat connectivity were concentrated in the downstream areas of the Han River basin. The spatial-distribution patterns of the wetlands seemed to follow a similar pattern. For the 101206 standard watershed, the normalized values of the aquatic-habitat connectivity and wetland were 0.00 (no reservoir) and 0.99, respectively, and the sub-index score of 0.90, which integrated the two normalized values, indicates a very healthy watershed. In contrast,  the normalized values of the aquatic-habitat connectivity and wetland for the 100201 standard watershed were 0.46 and 0.34, respectively, and the sub-index score of 0.28, which integrated the two normalized values, indicated an unhealthy watershed. At the 101801 standard watershed, the aquatic-habitat condition results from the aquatic-habitat connectivity (0.77) and wetland (0.66) indicators showed a relatively high value of 0.68.

A sub-index analysis of the TDI, BMI, and FAI (Figure 8) was conducted, except in the no-data areas (North Korea) in the Han River basin (Figure 12f).  The relationships of the TDI, BMI, and FAI were found to be significantly correlated. The TDI, BMI, and FAI were worse in the downstream areas.

However, the degree to which the TDI, BMI and FAI predict trophic diatom, benthic macroinvertebrate, and fish communities depends on the presence and levels of other stressors, such as large amounts of chlorophyll-a (Chl-a), low dissolved oxygen (DO) and biochemical oxygen (BOD), and high temperature. The normalized values of the TDI,

BMI and FAI were 0.70, 0.98, and 0.92, respectively, in the 101206 standard watershed located upstream; 0.69, 0.98, and 0.72, respectively, in the 100201 standard watershed located upstream; and 0.32, 0.25, and 0.25, respectively, in the 101801 standard watershed located downstream.

The sub-index scores after integrating the three normalized values were 0.91 and 0.83 for the 101206 and 100201 standard watersheds, respectively, indicating very healthy watersheds, and the sub-index score of 0.26 at the 101801 standard watershed indicated an unhealthy watershed.

The outputs of the watershed health provide basic data for local communities to proactively plan for growth. The sub-index results of the watershed-health assessment for each component can be optionally used to guide the master-planning process for watershed management at the watershed scale depending on the specific management objectives and can be combined with any of the other sub-indices in the Han River basin to determine priority conservation areas.

3.2 Assessment of the integrated watershed health

To assess the overall watershed health in the Han River basin, the results of the individual assessments were synthesized to provide an integrated watershed--health index score for the thirty-year period (1985–2014). The sample areas that were used to explain the differences in the watershed--health results for each component were the standard watersheds 101206 (urban 1.4% and forest 88.1%), 100201, (urban 0.8% and forest 88.2%), and 101801 (urban 9.8%

and forest 55.7%) (Figure 2a). The 101206, 100201, and 101801 standard watersheds were located in the upstream region of the Soyang dam (SYD), in the upstream region of the Chungju dam (CJD), and in the downstream region of the Paldang dam (PDD), respectively.

Figure 12 displays the normalized scores for each of the six attribute sub-indices and integrated watershed--health scores. The integrated watershed health exhibited a decreasinged tendency farther down the watershed. The integrated watershed health of the 101206 and 100201 standard watersheds was revealed to be very good, with ratings of 1 and

0.91, respectively. However, the 101206 standard watershed exhibited a distinctive weakness with respect to the hydrologic condition (0.06), especially in the surface (0.16) and groundwater (0.17). Although the 100201 standard watershed was a very healthy watershed, similar tolike the 101206 watershed, the formerit showed a distinctive weakness with respect to the water quality (0.1) and aquatic habitat condition (0.28). SIt is important to develop systematic plans must be developed to suit watershed circumstances and characteristics so that watershed management is more effective. The 101801 watershed was revealed to be in poor health, with a water--quality rating of 0.25. This area requires urgent action to restore the landscape, water quality, and biological conditions and to protect the water quantity. Table 5 shows the watershed--health scores in the test areas (upper/lower stream) of the Han River basin.

<Figure 12>

<Table 5>

**4. Conclusions**

In this study, a watershed--health assessment of the Han River basin in South Korea was performed by using monitoring data and SWAT modeling results. Six essential indicators of healthy watersheds were used in the assessment: 1) the landscape condition, 2) geomorphology, 3) hydrology, 4) water quality, 5) habitat, and 6) biological condition. In particular, athe sub-index of the watershed health that was related to the hydrology and water quality was developed to assess the possible long-term changes in the watershed by using SWAT modeling results.

During the most recent ten-year period (2005–2014), the watershed's health declined, as indicated by the worse results for the surface processes metric and soil water dynamics compared to those of the 1995–2004 period.

The spatial distributions of the poor watershed-health levels revealed the vulnerable areas in portionsparts of the CJD

watershed, upstream of the SYD watershed, and downstream of the PDD watershed with respect to the hydrology and water quality.

The sub-index results of the watershed-health assessment for each component can be used to guide the master-planning process for watershed management at the watershed scale based on specific management objectives and can be combined with any of the other sub-indices in the Han River basin tofor use in determineing priority conservation areas. Listing all the information of the watershed-health assessment can indicate vulnerable or healthy regions in the desired area and can provide basic data for action. The effectiveness of the watershed-health evaluation in this study can produce reliable information because this approach is entirely physically based. This approach can be utilized in a number of standard watersheds, local communities, and regions throughout the Han River basin and can be practically implemented in the watershed as a comprehensive watershed-management plan by government authorities or representative stakeholders.

Finally, the limitations of this study include the simulation of water quantity and quality data for possible long-term changes in the watershed model. Although the prediction of long-term water quantity and quality data with this modeling is essential to assess water-resource systems, the hydrologic and water quality conditions cannot be perfectly projected because of uncertainties in the models, climate data and other inputs that are required for the simulations. However, the results of this study are useful in terms of identifying potential watershed-health issues that are associated with ongoing watershed changes.

**Acknowledgments**

This research was supported by a grant (14AWMP-B082564-01) from the Advanced Water Management Research

Program funded by the Ministry of Land, Infrastructure and Transport of the Korean government.

[revised manuscript text omitted]

modeling for the period from 1985 to 2014 in the Han River basin.

[Figure]

Figure 7. Aquatic habitat conditions for the (a) aquatic habitat connectivity and (b) wetlands.

[Figure]

Figure 8. Biological conditions of the (a) FAI, (b) BMI and (c) FAI according to the observed monitoring data for the period from 2008 to 2013 in the Han River basin.

[Figure]

Figure 9. Change in hydrology and water quality for the (a) A (SYD watershed), (b) B (CJD watershed), (c) C (PDD

watershed), and (d) D (lower watershed) test areas for three ten-year periods.

[Figure]

Figure 10. WThe watershed--health index score changes for the hydrologic (a and b) and water quality (c and d)

conditions during the period 1995–2004 and the most recent ten-year period (2005–2014) based on the reference period (1985–1994).

Figure 11. PThe poor watershed health as revealed by the (a) hydrology, (b) water- quality, and (c) overlay results.

[Figure]

Figure 12. Watershed -health index _results_ for _the_ (a) landscape, (b) stream geomorphology, (c)

hydrology, (d) water quality, (e) aquatic habitat, (f) biological condition, and (g) integrated watershed health.

[Figure]

Table 1 Metrics and summary dataset that was used tofor the assess thement of watershed health in the study watershed.

| Component (metric) | Measurement method | Dataset |
|---|---|---|
| *Landscape* | | *GIS data* |
| Green infrastructure metric | Percentage of the watershed that is occupied by natural land cover | Land cover 2008[a] |
| Active river area metric | Percentage of natural land cover within the active river area | Land cover 2008, stream[b] |
| *Geomorphology* | | *GIS data* |
| Stream geomorphology metric | Percentage of assessed stream length in the reference condition | SRTM DEM (90×90)[c], stream |
| *Hydrology* | | *SWAT modeling data (1985–2014)* |
| Total metric | Precipitation and total runoff storage ratio | PREC, TQ |
| Surface processes metric | Surface runoff storage ratio | SQ |
| Soil water dynamics metric | Infiltration, soil water and lateral flow storage ratio | INFILT, SW, LQ |
| Groundwater dynamics metric | Percolation, groundwater recharge and return flow storage ratio | PERCOL, RECHARGE, GWQ |
| *Water quality* | | *SWAT modeling data (1985–2014)* |
| Water quality metric | Percentage of the assessed value in the reference criteria | Sediment, T-N, T-P |
| *Aquatic habitat condition* | | *GIS data* |
| Habitat connectivity metric | Reservoir density (number of reservoirs per stream length) | Reservoir location map[d], stream |
| Wetland metric | Percentage of the watershed that is occupied by wetlands | Land cover 2008 |
| *Biological condition* | | *Monitoring data (2008–2013)[e]* |
| Biological metric | Percentage of the assessed score in the reference condition | TDI, BMI, FAI |

Main data sources included [a] the Korea Ministry of Environment (KME); [b] the Ministry of Land, Infrastructure, and Transport (MOLIT) in
South Korea; [c] the International Center for Tropical Agriculture (CIAT); [d] the Korea Rural Community Corporation (KRC); and [e] the Korea
Ministry of Environment (KME) in South Korea (Ministry of Environment, 2013).

Table 2 Calibration and validation results for the dam inflow, dam-–storage volume, evapotranspiration and soil moisture, groundwater-–level fluctuation, sediments, T-N, and T-P at each calibration point.

| Model output | Evaluation criteria | Cal. | Val. | Cal. | Val. | Cal. | Val. | Cal. | Val. | Cal. | Val. | Cal. | Val. | Cal. | Val. |
|---|---|---|---|---|---|---|---|---|---|---|---|---|---|---|---|
| | Locations | HSD | | SYD | | CJD | | KCW | | YJW | | IPW | | PDD | |
| | $R^2$ | 0.82 | 0.84 | 0.90 | 0.89 | 0.81 | 0.74 | 0.90 | 0.63 | 0.91 | 0.62 | 0.93 | 0.59 | 0.92 | 0.88 |
| Dam inflow | NSE | 0.61 | 0.57 | 0.78 | 0.78 | 0.63 | 0.58 | 0.78 | 0.79 | 0.77 | 0.76 | 0.81 | 0.95 | 0.83 | 0.76 |
| (mm) | NSE (1/Q) | 0.44 | 0.26 | 0.49 | 0.56 | 0.34 | 0.25 | 0.47 | 0.60 | 0.46 | 0.47 | 0.62 | 0.75 | 0.65 | 0.51 |
| | RMSE (mm/day) | 7.9 | 9.3 | 3.8 | 3.9 | 3.5 | 3.1 | 6.5 | 0.7 | 9.1 | 2.4 | 9.2 | 2.9 | 0.8 | 2.3 |
| | PBIAS (%) | 14.5 | 12.5 | 10.3 | 14.0 | 8.9 | 9.9 | 18.0 | 4.9 | 25.5 | 14.1 | 25.6 | 17.2 | 2.2 | 6.8 |
| Dam storage | | HSD | | SYD | | CJD | | KCW | | YJW | | IPW | | PDD | |
| ($10^6$ m³) | $R^2$ | 0.73 | 0.77 | 0.94 | 0.96 | 0.87 | 0.84 | 0.57 | 0.85 | 0.47 | 0.83 | 0.47 | 0.79 | 0.40 | 0.44 |
| | PBIAS (%) | 18.9 | 9.9 | 16.3 | 9.3 | 18.2 | 15.2 | 5.1 | 7.4 | 3.7 | 11.1 | 9.1 | 7.2 | 0.9 | 1.4 |
| | Locations | SM | | CM | | - | | - | | - | | - | | - | |
| Evapotrans- | $R^2$ | 0.81 | 0.73 | 0.70 | 0.74 | - | - | - | - | - | - | - | - | - | - |
| piration (mm) | NSE | 0.64 | 0.45 | 0.50 | 0.55 | - | - | - | - | - | - | - | - | - | - |
| | RMSE (mm/day) | 2.3 | 9.1 | 4.0 | 3.0 | - | - | - | - | - | - | - | - | - | - |
| | PBIAS (%) | 9.6 | 30.2 | 11.6 | 23.7 | - | - | - | - | - | - | - | - | - | - |
| Soil moisture | Locations | SM | | CM | | - | | - | | - | | - | | - | |
| (%) | $R^2$ | 0.85 | 0.75 | 0.78 | 0.78 | - | - | - | - | - | - | - | - | - | - |
| Groundwater | Locations | - | | - | | GPGP | | YPGG | | YPYD | | YIMP | | HCGD | |
| level (EL.m) | $R^2$ | - | - | - | - | 0.70 | 0.63 | 0.64 | 0.45 | 0.70 | 0.41 | 0.53 | 0.40 | 0.69 | 0.67 |
| | Locations | SG | | CSG | | JW | | KCW | | YJW | | IPW | | PDD | |
| Sediment (tons) | $R^2$ | 0.78 | 0.70 | 0.78 | 0.76 | 0.90 | 0.71 | 0.54 | 0.64 | 0.84 | 0.54 | 0.69 | 0.66 | 0.72 | 0.80 |
| T-N (kg) | $R^2$ | 0.58 | 0.71 | 0.64 | 0.71 | 0.82 | 0.68 | 0.50 | 0.61 | 0.52 | 0.49 | 0.46 | 0.62 | 0.66 | 0.62 |
| T-P (kg) | $R^2$ | 0.77 | 0.77 | 0.88 | 0.88 | 0.80 | 0.56 | 0.56 | 0.58 | 0.50 | 0.47 | 0.66 | 0.70 | 0.74 | 0.69 |

[a] Cal. = calibration period (HSD, SYD, CJD and PDD: 2005-2009;, KCW, YJW and IPW: 2013) and Val. = validation period (HSD,
SYD, CJD and PDD: 2010-2014;, KCW, YJW and IPW: 2014)

Table 3 Description of the stream geomorphic conditions categories (Kline et al., 2009) and stream order for the watershed-health assessment of the geomorphic condition in the Han River basin.

| Condition | Description | River classification | Stream order (1–9) |
|---|---|---|---|
| Reference | In Equilibrium – no apparent or significant channel, floodplain, or land-cover modifications; the channel geometry is likely to be in balance with the flow and sediment that are produced in its watershed. | Mountainous river | 1 |
| Good | In Equilibrium but may be in transition into or out of the range of natural variability – minor erosion or lateral adjustment but adequate floodplain function; any adjustments from historical modifications nearly complete. | Small river | 2–3 |
| Fair | In Adjustment – moderate loss of floodplain function or moderate to major plan-form adjustments that could lead to channel avulsions. | Local river | 4–5 |
| Poor | In Adjustment and Stream Type Departure – may have changed to a new stream type, or central tendency of fluvial processes or significant channel and floodplain modifications may have altered the channel geometry such that the stream is not in balance with the flow and sediment that are produced in its watershed. | Urban river, National river | 6–9 |

Table 4 Summary of the hydrology, water–quality and biological criteria that were used to screen for the reference condition in the Han River basin.

| Component | Source | Reference condition |
|---|---|---|
| *Hydrology* | | |
| Precipitation | River–basin average of 30 years (1985–2014) as simulated by SWAT | 1,395.1 (mm) |
| Total runoff | | 919.5 (mm) |
| Surface runoff | | 249.4 (mm) |
| Infiltration | | 726.4 (mm) |
| Soil water storage | | 85.3 (mm) |
| Lateral flow | | 345.9 (mm) |
| Percolation | | 363.8 (mm) |
| Groundwater recharge | | 22.9 (mm) |
| Return flow | | 324.2 (mm) |
| *Water quality* | | |
| Sediment | LThe levels greater than the "marginally good" level on a seven-point scale | 15 (mg/L) |
| T-N | (excellent, very good, good, marginally good, fair, poor, very poor) of water– | 0.6 (mg/L) |
| T-P | quality criteria for streams and lakes as devised by the Basic Environmental Policy Act (BEPA) in South Korea. | 0.05 (mg/L) |
| *Biological condition* | | |
| TDI | The "Bbest" and "good" levels on a four-point scale (best, good, fair and | 72.5 |
| BMI | poor) of biological condition criteria devised by the Korea Ministry of | 80.0 |
| FAI | Environment (KME) (Ministry of Environment, 2013). | 78.1 |

Table 5 Watershed-health score results in each test area (upper/lower stream) of the Han River basin

| Component | A (SYD watershed) | B (CJD watershed) | C (PDD watershed) | D (Lower watershed) |
|---|---|---|---|---|
| *Landscape* | **0.80** | **0.66** | **0.53** | **0.26** |
| Green infrastructure metric | 0.85 | 0.67 | 0.52 | 0.25 |
| Active river area metric | 0.74 | 0.65 | 0.53 | 0.28 |
| *Geomorphology* | **0.75** | **0.47** | **0.46** | **0.54** |
| *Hydrology* | **0.21** | **0.74** | **0.37** | **0.60** |
| Total | 0.19 | 0.51 | 0.44 | 0.65 |
| Surface processes | 0.36 | 0.73 | 0.40 | 0.53 |
| Soil water dynamics | 0.61 | 0.44 | 0.58 | 0.39 |
| Groundwater dynamics | 0.30 | 0.55 | 0.45 | 0.58 |
| *Water quality* | **0.63** | **0.45** | **0.52** | **0.48** |
| Sediment | 0.40 | 0.29 | 0.55 | 0.61 |
| T-N | 0.76 | 0.70 | 0.49 | 0.32 |
| T-P | 0.52 | 0.40 | 0.53 | 0.53 |
| *Aquatic habitat condition* | **0.39** | **0.43** | **0.55** | **0.45** |
| Habitat connectivity | 0.22 | 0.30 | 0.52 | 0.40 |
| Wetland | 0.53 | 0.51 | 0.49 | 0.41 |
| *Biological condition* | **0.92** | **0.73** | **0.47** | **0.23** |
| TDI | 0.83 | 0.67 | 0.50 | 0.25 |
| BMI | 0.88 | 0.78 | 0.46 | 0.22 |
| FAI | 0.92 | 0.70 | 0.47 | 0.27 |
| *Integrated assessment* | **0.82** | **0.75** | **0.47** | **0.30** |

---

## Referee Report (RR1)

The authors have made substantial efforts to improve the manuscript and clearly addressed most of the points raised by both reviewers. I have only few minor comments to be addressed before final acceptance for publication.

1  "The calibrated parameters and hydrograph of the calibration results in the Han River basin were described by Chung et al (2017)." For calibrated parameter in Table 1.( Descriptions of calibrated parameters in SWAT), how can we relate these influential parameters to physical process of hydrology? Some discussion may need to be added.

2 How to explain the differences between the adjusted values for dams and weirs? For example, the adjusted values for reservoir related parameters, RES_ESA, RES_EVOL, RES_PSA and so on are different for dams and weirs. How to interpolate these differences? Again, it is better to relate to physical process.

3 The calibrated values are average values in Table 1. Then did you use these average values across the entire study area?